# Mitigating Privacy–Utility Trade-off in Decentralized Federated Learning via $f$-Differential Privacy

**Xiang Li**[*]
University of Pennsylvania
lx10077@upenn.edu

**Buxin Su**[*]
University of Pennsylvania
subuxin@upenn.edu

**Chendi Wang**[*†]
Xiamen University
University of Pennsylvania
chendi.wang@xmu.edu.cn

**Qi Long**[†]
University of Pennsylvania
qlong@upenn.edu

**Weijie Su**[†]
University of Pennsylvania
suw@wharton.upenn.edu

## Abstract

Differentially private (DP) decentralized Federated Learning (FL) allows local users to collaborate without sharing their data with a central server. However, accurately quantifying the privacy budget of private FL algorithms is challenging due to the co-existence of complex algorithmic components such as decentralized communication and local updates. This paper addresses privacy accounting for two decentralized FL algorithms within the $f$-differential privacy ($f$-DP) framework. We develop two new $f$-DP–based accounting methods tailored to decentralized settings: Pairwise Network $f$-DP (PN-$f$-DP), which quantifies privacy leakage between user pairs under random-walk communication, and Secret-based $f$-Local DP (Sec-$f$-LDP), which supports structured noise injection via shared secrets. By combining tools from $f$-DP theory and Markov chain concentration, our accounting framework captures privacy amplification arising from sparse communication, local iterations, and correlated noise. Experiments on synthetic and real datasets demonstrate that our methods yield consistently tighter $(\epsilon, \delta)$ bounds and improved utility compared to Rényi DP–based approaches, illustrating the benefits of $f$-DP in decentralized privacy accounting.

## 1 Introduction

Federated Learning (FL) has emerged as a powerful paradigm for privacy-preserving machine learning, enabling user devices to collaboratively train models without sharing raw data [48, 31]. In scenarios where a central server is unavailable, untrusted, or communication is expensive, decentralized FL becomes a compelling alternative. Algorithms such as decentralized SGD [42, 41] offer scalability and low communication overhead, making them well-suited for edge computing and peer-to-peer applications [52, 69, 56, 37, 59, 34, 64, 62].

Despite its decentralized nature, FL remains susceptible to privacy leakage. Even without access to raw data, model updates can reveal sensitive information [55, 67], especially under inference or reconstruction attacks [50, 27]. Differential Privacy (DP) [19] offers a formal and principled solution by bounding the effect of any individual data on model outputs. A widely used approach is differentially private stochastic gradient descent (DP-SGD), which adds noise to stochastic gradients to ensure privacy [1, 7]. In general, more accurate privacy accounting, i.e., quantifying cumulative

---

[*]Equal contribution.
[†]Corresponding authors.

39th Conference on Neural Information Processing Systems (NeurIPS 2025).

privacy loss over multiple training rounds, allows for adding less noise to achieve the same privacy guarantee, thereby improving utility. However, in decentralized settings—where users communicate over graphs, activate randomly, and share information sparsely—privacy accounting remains a significant challenge [44, 30, 47, 21].

A key factor influencing the difficulty of privacy accounting is the choice of privacy notion, which governs how adversarial capabilities are modeled and how much noise must be added to ensure privacy. Several existing decentralized DP-SGD protocols adopt strong privacy notions such as local differential privacy (Local DP) [22, 33, 63, 43, 46], which assume adversaries can observe all local updates. While offering strong protection, Local DP requires injecting substantial noise, significantly degrading model performance. To address this limitation, recent work has proposed relaxed notions like Pairwise Network DP (PN-DP) [16, 15], which better capture the partial observability inherent in decentralized systems. In parallel, the Secret-based Local DP (SecLDP) framework [2, 60] considers scenarios where user pairs conspire and share secrets with their neighbors and coordinate to add correlated noise to their updates, enabling improved utility-privacy trade-offs under limited collusion.

However, existing analyses of both PN-DP and SecLDP rely on Rényi differential privacy (RDP) [49] or $(\epsilon, \delta)$-DP, whose privacy bounds are often loose—especially for iterative algorithms. Given the prevalence of iterative algorithms such as DP-SGD, we ask: Is it possible to develop a fine-grained privacy accounting framework for decentralized FL that consistently improves the privacy–utility trade-off across different algorithmic designs?

**Contribution.** Our answer is affirmative: we show that $f$-differential privacy ($f$-DP) [17] provides a unified and effective analytical framework for privacy accounting in decentralized FL. We summarize our contribution below.

1. **New $f$-DP notions for decentralized FL.** To enable tight privacy accounting, we introduce two $f$-DP–based notions tailored to decentralized settings: (1) *Pairwise Network $f$-DP (PN-$f$-DP)* extends PN-DP to quantify fine-grained privacy loss between user pairs in random walk communication, supporting both user-level (Theorem 4.1) and record-level (Theorem E.1) guarantees; (2) *Secret-based $f$-Local DP (Sec-$f$-LDP)* generalizes $f$-DP to regimes with correlated noise and shared secrets, as in DecoR-style algorithms (Theorem 4.2).

2. **Refined privacy guarantees via $f$-DP.** To illustrate, we analyze two representative and practical variants of DP-SGD: (1) *Decentralized DP-SGD with random-walk communication* (Algorithm 1), where updates propagate via a random walk in a communication graph [15]; and (2) *DP-SGD with correlated noise* (Algorithm 2), where user pairs inject structured noise via shared secrets [2]. For *random-walk communication*, we develop a PN-$f$-DP analysis that captures privacy amplification from: (i) *communication sparsity*, (ii) *local iteration updates*, and (iii) *Markov hitting times*. Our approach leverages the joint concavity of trade-off functions and Markov concentration to yield tighter $(\epsilon, \delta)$ bounds than RDP. For *correlated-noise methods*, we extend $f$-DP via Sec-$f$-LDP to account for secret sharing and partial trust, producing sharper bounds under limited collusion.

3. **Improved privacy–utility trade-offs.** Empirical results on both real and synthetic datasets show that $f$-DP–based accounting consistently yields tighter $(\epsilon, \delta)$ bounds than existing methods, requiring less noise for a given privacy level and leading to significantly improved model utility.

## 1.1 Related works

**DP notions in FL.** Local DP assumes all local models are observable by potential attackers [22, 33, 63, 43, 45]. However, this strong observability assumption often requires injecting substantial noise, which severely degrades utility in practice. To mitigate this issue, Pairwise Network DP (PN-DP) was recently proposed [16, 15]. PN-DP relaxes the Local DP assumption by modeling only pairwise observability—attackers can access the local model involved in each decentralized communication round, rather than all local models. The prefix "PN" reflects that data exchange occurs pairwise between connected nodes in a network. This relaxation enables the privacy amplification by decentralization [14], providing more fine-grained DP guarantees and improving utility in decentralized learning. Building upon this line of work, our PN-$f$-DP framework extends PN-DP by adopting the $f$-DP formalism [17], which provides lossless privacy accounting through a hypothesis testing interpretation. Existing PN-DP analyses rely mainly on Rényi DP (PN-RDP) [49], which often yields loose bounds, while $f$-DP achieves tighter characterization. Furthermore, unlike previous federated $f$-DP formulations [70], our PN-$f$-DP captures the additional privacy amplification arising

from decentralization, random walks, and iterative communication—crucial algorimithic components in realistic networked settings.

In some decentralized scenarios, users may further collude and share secrets [54, 2, 60], creating correlations in their local randomness. This behavior is formalized as Secret-based Local DP (SecLDP) by [2], which allows the injection of correlated noise to align the privacy–utility trade-off with that of central DP. Extending this idea, our Sec-$f$-LDP framework leverages shared secrets to incorporate correlated noise, thereby achieving additional privacy amplification in structured collusion scenarios—an aspect not addressed by standard Local DP.

**Privacy amplification in FL.** The privacy analysis of decentralized DP-SGD is intricate due to potential privacy amplification from randomized data processing. Privacy amplification refers to techniques that enhance privacy protection by reducing the amount of information an attacker can infer from the output. It can result from both shuffling [13, 24, 25, 35, 61] and decentralization [14, 16, 15]. Beyond these, the latest privacy analysis of decentralized DP-SGD by [15] incorporates the effects of iterative processes and random walks. In the case of iterative processes, only the final model parameter is revealed, not the intermediate ones, which influences the random noise applied to gradients, leading to *privacy amplification by iteration* [26]. This phenomenon has been further explored in DP-SGD [3, 68] and extended into the $f$-DP framework via a shifted interpolated process [8]. The iterative process is closely related to the intermittent communication in FL, where multiple local updates occur between consecutive communication rounds without revealing any information. On the other hand, the privacy impact of random walks has only been studied under RDP [15], and it remains unclear how to capture this effect within the finer $f$-DP framework.

## 2 Preliminaries

### 2.1 Preliminaries on differential privacy

Differential Privacy [19] is a formal framework designed to protect individual data records by ensuring that the output of a randomized algorithm remains nearly unchanged when a single entry in the dataset is modified. Formally, let $D = \{z_i\}_{i=1}^n \subset \mathcal{Z}$ be a dataset of size $n$ drawn from some data domain $\mathcal{Z}$. The classical definition of $(\epsilon, \delta)$-DP is as follows:

**Definition 2.1** ($(\epsilon, \delta)$-DP)**.** A randomized algorithm $\mathcal{A}$ satisfies $(\epsilon, \delta)$-DP if, for all neighboring datasets $D$ and $D'$ differing in at most one entry, and for any measurable set $S \subseteq \mathcal{S}$, it holds that

$$\mathbb{P}[\mathcal{A}(D) \in S] \le e^\epsilon \mathbb{P}[\mathcal{A}(D') \in S] + \delta.$$

Intuitively, small values of $\epsilon$ and $\delta$ imply that the distributions of $\mathcal{A}(D)$ and $\mathcal{A}(D')$ are nearly indistinguishable. As a result, the presence or absence of any individual data point has limited influence on the algorithm's output, thereby preserving privacy. Rényi Differential Privacy (RDP) [49] extends this notion by measuring privacy loss using the Rényi divergence between $\mathcal{A}(D)$ and $\mathcal{A}(D')$. This formulation enables more precise privacy accounting under composition, often yielding cleaner and tighter bounds than standard composition results for $(\epsilon, \delta)$-DP.

**Definition 2.2** (Rényi DP [49])**.** A randomized mechanism $\mathcal{A}$ satisfies $(\alpha, \epsilon)$-Rényi DP ($(\alpha, \epsilon)$-RDP) if, for $D$ and $D'$ that differ in one element, we have $R_\alpha(\mathcal{A}(D) \| \mathcal{A}(D)) \le \epsilon$, for all $\alpha > 1$, where $R_\alpha(P\|Q) := \frac{1}{\alpha-1} \log \int \left(\frac{p(x)}{q(x)}\right)^\alpha q(x) dx$ is the Rényi divergence between distributions $P$ and $Q$.

Besides different divergences, the distinguishability between $\mathcal{A}(D)$ and $\mathcal{A}(D')$ can be measured using the hypothesis testing formulation [66, 32, 58] and is systematically studied as $f$-DP by [17]. More specifically, consider a hypothesis testing problem $H_0 : \text{data} \sim P$ versus $H_1 : \text{data} \sim Q$ and a rejection rule $\phi \in [0, 1]$. We define the type I error as $\alpha_\phi = \mathbb{E}_P[\phi]$, which represents the probability of mistakenly rejecting the null hypothesis $H_0$. The type II error, $\beta_\phi := 1 - \mathbb{E}_Q[\phi]$, is the probability of incorrectly accepting the alternative hypothesis $H_1$. The trade-off function $T(P, Q)$ denotes the minimal type II error at level $\alpha$ of type I error, expressed as: $T(P, Q)(\alpha) = \inf_\phi \{\beta_\phi : \alpha_\phi \le \alpha\}$.

**Definition 2.3** ($f$-DP and Gaussian DP [17])**.** A mechanism $\mathcal{A}$ is said to satisfy $f$-DP if for any datasets $D$ and $D'$ that differ in one element, the inequality $T(\mathcal{A}(D), \mathcal{A}(D')) \ge f$ holds pointwisely. In particular, $\mathcal{A}$ satisfies $\mu$-Gaussian DP ($\mu$-GDP) if it is $G_\mu$-DP with $G_\mu(x) = \Phi(\Phi^{-1}(1 - x) - \mu)$, where $\Phi$ denotes the cumulative distribution function (cdf) of the standard normal distribution.

A mechanism satisfying $f$-DP is considered more private when its associated trade-off function $f$ is larger. In the extreme case where $\mathcal{A}(D)$ and $\mathcal{A}(D')$ are indistinguishable, the trade-off function attains its maximum: the identity function $\mathrm{Id}(x) := 1 - x$. Therefore, any valid trade-off function must satisfy $f \leq \mathrm{Id}$ pointwise. A trade-off function $f = T(P, Q)$ is said to be symmetric if $T(P, Q) = T(Q, P)$. As shown by [17], any trade-off function can be symmetrized. Throughout this paper, we assume all trade-off functions are symmetric unless stated otherwise. Finally, the composition of DP mechanisms admits a natural interpretation in $f$-DP: it corresponds to the tensor product of trade-off functions (see Def. 2.4).

**Definition 2.4** (Tensor product of trade-off functions [17]). For two trade-off functions $f = T(P, Q)$ and $f' = T(P', Q')$, the tensor product between $f$ and $f'$ is defined as $f \otimes f' = T(P \times P', Q \times Q')$. Specifically, the $n$-fold tensor product of $f$ itself is denoted as $f^{\otimes n}$.

## 2.2 Pairwise network differential privacy

**Communication graph.** In decentralized FL, we assume that communications take place through a connected communication graph $\mathcal{G} = (V, E)$, where $V = \{1, \cdots, n\}$ represents a set of $n$ vertices (or nodes). Each node $i \in V$ corresponds to a local user, and every local user $i$ holds a local dataset $D_i$. Collectively, these datasets are represented as $D = \cup_{i=1}^{n} D_i$. Communication between users is facilitated by edges: if an edge $(i, j) \in E$ exists, users $i$ and $j$ can exchange information. To model these interactions, we introduce a transition matrix $W \in \mathbb{R}^{n \times n}$, which is a Markov matrix satisfying $\sum_{j=1}^{n} W_{ij} = 1$ for any $i \in [n]$. Each entry $W_{ij}$ reflects the probability of transmitting a message (such as model updates) from node $i$ to node $j$ in the next step. Specifically, for the communication graph $(V, E)$, we have $W_{ij} > 0$ if $(i, j) \in E$, and $W_{ij} = 0$ otherwise.

**Two DP levels.** We consider two DP notions with different granularities in decentralized learning: **user-level DP**, which bounds the influence of any single user's entire dataset on the algorithm's output [40, 28], and **record-level DP**, which limits the impact of changing a single data point within a user's dataset [70, 12]. In user-level DP, datasets $D$ and $D'$ are adjacent (denoted $D \sim D'$ or $D \sim_i D'$) if they differ in all records of a single user $i$, offering stronger privacy. In contrast, record-level DP defines adjacency (denoted $D \approx D'$ or $D \approx_i D'$) based on a difference in just one record. User-level DP provides stronger protection but lacks certain benefits like privacy amplification by sub-sampling, which can enhance record-level DP guarantees but does not apply when entire user datasets differ.

**Pairwise network differential privacy (PN-DP).** PN-DP is a recent relaxation of local DP tailored for decentralized FL [15]. It applies to settings where user $j$ has limited visibility of the overall algorithm output. PN-DP quantifies the privacy leakage from user $i$'s data to user $j$ by bounding the distinguishability of $\mathcal{A}_j(D)$ and $\mathcal{A}_j(D')$, where $D \sim_i D'$ are user-level adjacent datasets. Here, $\mathcal{A}_j(D)$ denotes user $j$'s view of the algorithm's output. Existing work focuses on user-level PN-DP using RDP tools [14, 16, 15]. In Section 3, we develop our own PN-$f$-DP framework and also incorporate record-level analysis for finer privacy guarantees.

**Definition 2.5** (User-level pairwise network RDP, or PN-RDP [16]). For a function $\epsilon : V \times V \to \mathbb{R}^+$, an algorithm $\mathcal{A}$ satisfies $(\alpha, \epsilon)$-pairwise network RDP if for all pairs $(i, j) \in V \times V$ and for two datasets $D \sim_i D'$, $R_\alpha(\mathcal{A}_j(D) \| \mathcal{A}_j(D')) \leq \epsilon(i, j)$.

## 2.3 Private Decentralized SGD with local updates

We study two representative algorithms in decentralized differentially private learning: decentralized DP-SGD (Algorithm 1) and DecoR (Algorithm 2).

Algorithm 1 describes decentralized DP-SGD with local updates. At each communication round $t$, the active user $i_t$ performs $K$ local SGD steps using mini-batch gradients $g_{k,t}$, each perturbed with Gaussian noise $Z_{k,t} \sim \mathcal{N}(0, \sigma^2)$ scaled by the $\ell_2$-sensitivity $\Delta$. These updates are optionally projected onto a convex set $\mathcal{K}$ via $\Pi_{\mathcal{K}}$, though we focus on the unconstrained case. After $K$ steps, the model $\theta_{K,t}$ is sent to a new user $j \sim W_{i_t}$ based on the random walk transition matrix $W$, and the process continues with $i_{t+1} = j$. We analyze the general case of $K \geq 1$, extending prior work focused on $K = 1$ [15].

Algorithm 2 (DECOR) describes a parallel variant where each user $i$ clips its gradient to norm $\Delta$, adds structured noise—correlated noise shared with neighbors ($Z_{ij,t} = -Z_{ji,t}$) and independent noise $\bar{Z}_{i,t} \sim \mathcal{N}(0, \sigma_{\mathrm{DP}}^2 I_d)$—and updates its model. Users then average their intermediate models via a mixing matrix $W$. This design enables a decentralized implementation of SecLDP [2], improving privacy–utility trade-offs under limited trust assumptions.

| **Algorithm 1:** Decentralized DP-SGD | **Algorithm 2:** DECOR: DP-SGD with Corr. Noise |
|---|---|

**Algorithm 1:** Decentralized DP-SGD

**Input:** Transition matrix $W$, communication rounds $T$, $K$, $\mathcal{K}$, start node $i_0$; init $\theta_{0,0}$, stepsizes $\eta$, batch size $b$, sensitivity $\Delta$, $\sigma$, $\{\ell_i\}_{i \in V}$

**for** $t = 0$ **to** $T-1$ **do**
   **for** $k = 0$ **to** $K-1$ **do**
      Sample mini-batch $g_{k,t}$ with $\mathbb{E}[g_{k,t}] = \nabla \ell_{i_t}(\theta_{k,t})$;
      Sample $Z_{k,t} \sim \mathcal{N}(0, \sigma^2)$;
      Update $\theta_{k+1,t} \leftarrow \Pi_{\mathcal{K}}(\theta_{k,t} - \eta(g_{k,t} + Z_{k,t}))$;
   **end**
   Sample $j \sim W_{i_t}$;
   Send $\theta_{K,t}$ to $j$;
   Set $i_{t+1} \leftarrow j$;
**end**
**Output:** $\theta_{K,T}$

**Algorithm 2:** DECOR: DP-SGD with Corr. Noise

**Input:** User $i$ initializes $\theta_{i,0}$; transition matrix $W$; communication rounds $T$, stepsizes $\eta$, sensitivity $\Delta$, $\sigma_{\text{cor}}$, $\sigma_{\text{DP}}$, $\{\ell_i\}$

**for** $t = 0$ **to** $T-1$ **do**
   **for** $i = 1$ **to** $n$ **do**    // in parallel
      Sample a data point and compute $g_{i,t} := \text{Clip}(\nabla \ell_i(\theta_{i,t}), \Delta)$;
      **for** $j \in \mathcal{N}_i$ **do**    // neighbors of $i$
         Sample $Z_{ij,t} = -Z_{ji,t} \sim \mathcal{N}(0, \sigma_{\text{cor}}^2 I_d)$;
      **end**
      $\tilde{g}_{i,t} := g_{i,t} + \sum_{j \in \mathcal{N}_i} Z_{ij,t} + \mathcal{N}(0, \sigma_{\text{DP}}^2 I_d)$;
      $\theta_{i,t+\frac{1}{2}} \leftarrow \theta_{i,t} - \eta \tilde{g}_{i,t}$;
      $\theta_{i,t+1} \leftarrow \sum_j W_{ij} \theta_{j,t+\frac{1}{2}}$;
   **end**
**end**
**Output:** $\theta_{K,T}$

# 3 $f$-Differential Privacy Notions for Decentralized Learning

To obtain tighter privacy guarantees for Algorithm 1, we introduce Pairwise Network $f$-Differential Privacy (PN-$f$-DP). We first focus on user-level privacy, where two datasets $D$ and $D'$ are adjacent at the user level ($D \sim_i D'$). Let $\mathcal{A}(D)$ denote the output of Algorithm 1 on dataset $D$, and let $\mathcal{A}_j(D)$ represent user $j$'s view—i.e., all intermediate messages received by $j$ during training.

**Definition 3.1** (User-level PN-$f$-DP). Let $f : V \times V \times [0,1] \to [0,1]$ be such that $f_{ij} := f(i, j, \cdot)$ is a trade-off function for any $i, j \in V$. A decentralized algorithm $\mathcal{A}(D) = (\mathcal{A}_j(D))_{j \in V}$ satisfies user-level pairwise network $f$-differential privacy if $T(\mathcal{A}_j(D), \mathcal{A}_j(D')) \geq f_{ij}$ for all nodes $i, j \in V$ and two user-level neighboring datasets $D \sim_i D'$.

We similarly define the record-level variant, where $D \approx_i D'$ denotes datasets differing in exactly one record held by user $i$.

**Definition 3.2** (Record-level PN-$f$-DP). Let $f : V \times V \times [0,1] \to [0,1]$ be such that $f_{ij} := f(i, j, \cdot)$ is a trade-off function for any $i, j \in V$. A decentralized algorithm $\mathcal{A}(D) = (\mathcal{A}_j(D))_{j \in V}$ satisfies record-level pairwise network $f$-differential privacy if $T(\mathcal{A}_j(D), \mathcal{A}_j(D')) \geq f_{ij}$ for all nodes $i, j \in V$ and two record-level neighboring datasets $D \approx_i D'$.

**Remark 3.1.** Our definition of record-level PN-$f$-DP is a relaxation of weak federated $f$-DP in [70]. Specifically, if a mechanism $\mathcal{A}$ satisfies record-level PN-$f$-DP, then it also satisfies weak federated $\tilde{f}$-DP, where $\tilde{f} = (\min_{i,j} f_{ij})$ with $(\cdot)$ the double convex conjugate operator. Note that for a function $g : \mathbb{R} \to \mathbb{R}$, its convex conjugate is defined by $g^*(y) := \max_x \{xy - g(x)\}$.

In some decentralized learning scenarios, a pair of users $(i, j)$ may share a sequence of secrets $\mathbf{S}_{ij}$. Privacy analysis can be performed by relaxing local differential privacy and introducing the concept of SecLDP, as discussed in [2]. Users with secrets may injected correlated noise before gossip averaging and use uncorrelated noise to protect the gossip average, as shown in Algorithm 2 proposed by [2].

**Definition 3.3** (Sec-$f$-LDP). Let $\mathcal{S}_\mathcal{I} = (\mathbf{S}_{ij}, i, j \in \mathcal{V})$ be a set of secrets with some indices $\mathcal{I} \subset \mathcal{V}$. We say a randomized decentralized algorithm $\mathcal{A}$ satisfies $\mathcal{S}_\mathcal{I}$-Sec-$f$-LDP if $T(\mathcal{A}(D)|\mathcal{S}_\mathcal{I}$ is hidden, $\mathcal{A}(D')|\mathcal{S}_\mathcal{I}$ is hidden$) \geq f$, for any neighboring datasets $D, D'$. Specifically, if $f = T(\mathcal{N}(0, 1), \mathcal{N}(\mu, 1))$, we say $\mathcal{A}$ satisfies $\mathcal{S}_\mathcal{I}$-Sec-$\mu$-GLDP (secret Gaussian local differential privacy).

# 4 Privacy Analysis

This section presents the privacy guarantees for the two proposed algorithms. For Algorithm 1, Section 4.1 analyzes the user-level privacy guarantees, while Section 4.2 focuses on the record-level ones. For Algorithm 2, Section 4.3 investigates the $f$-DP characterization under correlated noise.

## 4.1 User-level Privacy Analysis

The privacy analysis of Algorithm 1 is divided into two steps, following the spirit in [15]. In the first step, we bound the privacy loss incurred each time the algorithm visits node $j$. To obtain a tighter bound, we leverage $f$-DP techniques that account for the mixture mechanisms and iterative structure induced by the random walk and updates in Algorithm 1. In the second step, we analyze the total number of node visits using a recently developed Hoeffding-type inequality for Markov chains [23]. The overall privacy loss is then obtained by composing the per-visit guarantees across all visits. A detailed proof is provided in Appendix D.

**Mixture distributions from random walks.** We now present the first step of our analysis. Consider a random variable, $\mathcal{A}_j^{\text{single}}(D)$, which represents the model first observed by the user $j$ after it has been updated using data from some node $i$. Let $\mathcal{A}_j^t(D)$ denote the model observed by user $j$ at time $t$ for the first time. By the properties of random walks, the probability that the model reaches node $j$ from node $i$ for the first time at time $t$ is given by $w_{ij}^t := \mathbb{P}[\tau_{ij} = t]$, where $\tau_{ij}$ is the hitting time from $i$ to $j$. If $\tau_{ij} \geq T+1$, then user $j$ does not observe the model within $T$ steps, ensuring perfect privacy: $\mathcal{A}_j^{T+1}(D) = \mathcal{A}_j^{T+1}(D')$ for all $D \sim_i D'$. We define $w_{ij}^{T+1} := \mathbb{P}[\tau_{ij} \geq T+1] = 1 - \sum_{t=1}^T w_{ij}^t$ as the probability that the model is not observed by node $j$ within $T$ steps. Hence, the distribution of the one-time snapshot $\mathcal{A}_j^{\text{single}}(D)$ can be expressed as a mixture over $\{\mathcal{A}_j^t(D)\}_{t=1}^{T+1}$, with weights $\{w_{ij}^t\}_{t=1}^{T+1}$ reflecting the timing of the first visit.

To analyze the privacy of the mixture mechanism $\mathcal{A}_j^{\text{single}}(D)$, we leverage the joint concavity of trade-off functions established in [61]. Let $P_j^t$ and $Q_j^t$ denote the distributions of $\mathcal{A}_j^t(D)$ and $\mathcal{A}_j^t(D')$, with respective densities $p_j^t$ and $q_j^t$, and define the corresponding trade-off function as $f_{ij}^t := T(P_j^t, Q_j^t)$ for $D \sim_i D'$. The key insight is that the overall privacy loss—despite arising from a random and time-dependent mixture—can be lower bounded by a convex combination of the per-step trade-offs $f_{ij}^t$'s, weighted by the same weight $\{w_{ij}^t\}_{t=1}^{T+1}$. This is formalized in the following lemma.

**Remark 4.1.** The idea of focusing on a single observation $\mathcal{A}_j^{\text{single}}(D)$ is inspired by [15]. However, unlike their approach, which defines the weights $w_{ij}^t$ as the matrix entries $(W^t)_{ij}$, we instead define $w_{ij}^t$ based on the hitting time distribution. This subtle shift better captures the random walk dynamics and leads to tighter bounds. As shown in our experimental results in Section 5, converting our $f$-DP guarantees back into RDP yields improved bounds, primarily due to the more accurate modeling of communication timing via hitting times.

**Lemma 4.1.** *For any $D \sim_i D'$, we have $T(\mathcal{A}_j^{\text{single}}(D), \mathcal{A}_j^{\text{single}}(D')) \geq f_{ij}^{\text{single}}$, where $f_{ij}^{\text{single}}$ is defined as follows: for all $s \in [0, \infty)$,*

$$f_{ij}^{\text{single}}(\alpha(s)) = \sum_{t=1}^T w_{ij}^t f_{ij}^t(\alpha_t(s)) + w_{ij}^{T+1}(1 - \alpha_{T+1}(s)),$$

*with $\alpha_t(s) = \Pr_{X \sim P_j^t}\left[\frac{p_j^t(X)}{q_j^t(X)} > s\right]$ for any $t$, $\alpha_{T+1}(s) = \mathbf{1}_{[s<1]}$, and $\alpha(s) = \sum_{t=1}^{T+1} w_{ij}^t \alpha_t(s)$.*

**Remark 4.2.** The joint concavity (lower) bound $f_{ij}^{\text{single}}$ for trade-off functions in Lemma 4.1 is widely used in $f$-DP literature, and [61] provides necessary and sufficient conditions under which this bound is tight. In general, the tightness depends on the behavior of the likelihood ratio of the mixture, which in our case is determined by a complex, data-dependent walk over the graph topology.

**Computation of the weights $w_{ij}^t$.** A remaining issue is how to compute the weights $w_{ij}^t$. Recall that $w_{ij}^t = \mathbb{P}[\tau_{ij} = t]$, which denotes the probability that user $j$ observes the model for the first time at the $t$-th iteration. To compute $w_{ij}^t$ recursively, we begin with the base case: for $t = 1$, the model is transmitted directly from user $i$ to user $j$, so $w_{ij}^1 = W_{ij}$. For $t > 1$, we decompose the event $\{\tau_{ij} = t\}$ based on the first step of the random walk. If the model is sent to a user $k \neq j$ in the first step, the event $\{i_1 = k, \tau_{ij} = t\}$ is equivalent to $\{\tau_{kj} = t-1\}$, meaning the model must reach $j$ from $k$ in exactly $t-1$ steps, without visiting $j$ beforehand. By the Markov property, we obtain the recurrence: $w_{ij}^t = \sum_{k \neq j} W_{ik} w_{kj}^{t-1}$. The initialization satisfies $w_{ii}^0 = 1$ and $w_{ij}^0 = 0$ for $j \neq i$, as the model starts at user $i$.

**Computation of individual trade-off $f_{ij}^t$.** To compute each trade-off function $f_{ij}^t$, we apply the composition rule of $f$-DP, which accounts for the cumulative privacy effect of local updates. Since noise is added at each step, these updates offer privacy protection. When the loss functions are strongly convex, we apply privacy amplification by iteration in $f$-DP [8], which leverages both the contraction effect and repeated noise injection to yield tighter trade-off functions. In contrast, for general non-convex loss functions, we use the standard composition rule from Def. 2.4, resulting in a more conservative privacy bound. The following lemmas summarize both cases.

**Lemma 4.2.** *Assume that the loss functions $\ell_i$ are $m$-strongly convex and $M$-smooth. Let $c = \max\{|1 - \eta m|, |1 - \eta M|\}$. Then, for $0 < c < 1$, we have $f_{ij}^t \geq G_{\mu_t}$ with $\mu_t = \sqrt{c^{2K(t-1)} \cdot \frac{1+c}{1-c} \cdot \frac{(1-c^K)^2}{1-c^{2Kt}}} \frac{\Delta}{\sigma}$. When $c = 1$, one has $\mu_t = \frac{\sqrt{K}\Delta}{\sigma\sqrt{tK+1}}$.*

**Lemma 4.3.** *For non-convex loss functions with gradient sensitivity $\Delta$, we have $f_{ij}^t \geq G_{\frac{\sqrt{K}\Delta}{\sqrt{tK+1}\sigma}}$.*

**Final privacy guarantee: Combining all components together.** With $f_{ij}^{\text{single}}$ from Lemma 4.1 and a lower bound on each $f_{ij}^t$ from Lemma 4.2 or 4.3, we can compute a valid lower bound on the trade-off function $T(\mathcal{A}_j^{\text{single}}(D), \mathcal{A}_j^{\text{single}}(D')) \geq f_{ij}^{\text{single}}$. This bounds the privacy leakage from user $i$ to user $j$ at the time when the model is first observed by $j$. Since Algorithm 1 repeatedly updates and transmits the model, this process may occur multiple times. Thus, it suffices to compose the per-visit trade-off function $f_{ij}^{\text{single}}$ according to the number of visits to user $j$. To bound this number, we apply a concentration inequality for Markov chains [23]. Details are deferred in the Appendix.

Combining these components yields the final privacy guarantee in Theorem 4.1. Although it lacks a closed-form expression, the bound can be efficiently computed numerically.

**Theorem 4.1.** *Assume the transition matrix $W$ is irreducible, aperiodic, and symmetric, with a spectral gap $1 - \lambda_2 > 0$, where $\lambda_2$ is the second-largest eigenvalue of $W$ (see Appendix A for definitions). Then, for $D \sim_i D'$, under the assumptions in Lemma 4.2 (strongly convex case) or Lemma 4.3 (non-convex case), we have*

$$T(\mathcal{A}_j(D), \mathcal{A}_j(D')) \geq \left(f_{ij}^{\text{single}}\right)^{\otimes \lceil (1+\zeta)T/n \rceil}$$

*with probability $1 - \delta'_{T,n}$, for any $\zeta > 0$ and $\delta'_{T,n} = \exp\left(-\frac{1-\lambda_2}{1+\lambda_2} \cdot 2\zeta^2 T/n^2\right)$. The probability is taken over the randomness of the random walk initialized from the stationary distribution.*

**Remark 4.3.** Lemma 4.1 holds for any valid trade-off function $f_{ij}^t$. In this work, we focus on the Gaussian lower bound (in Lemmas 4.2 or 4.3) for its simplicity and (asymptotic) universality [18]. Consequently, Theorem 4.1 also applies to general lower bounds of trade-off functions, provided that they are valid and their mixture distributions can be efficiently computed.

**Remark 4.4.** The slack term $\delta'_{T,n}$ accounts for uncertainty in the number of times a user $j$ is visited during training [15]. With probability at least $1 - \delta'_{T,n}$, user $j$ is visited no more than $\lceil (1 + \zeta)T/n \rceil$ times, yielding the composed trade-off function $(f_{ij}^{\text{single}})^{\otimes \lceil (1+\zeta)T/n \rceil}$ in Theorem 4.1. If the exact visit count $N$ were known, the bound $(f_{ij}^{\text{single}})^{\otimes N}$ would apply instead. Thus, $\delta'_{T,n}$ merely captures uncertainty from concentration bounds without materially affecting the results.

## 4.2 Extension to Record-level Privacy

Our analysis naturally extends to the record-level setting, which provides finer-grained privacy by protecting individual data records rather than entire users. In this case, each user computes a stochastic gradient using a random subset of their local data, which reduces the probability that any given record is selected—thus amplifying privacy. Technically, this effect can be captured using privacy amplification by sub-sampling [33, 5, 65, 72, 71]. We consider two datasets $D$ and $D'$ that differ by a single record belonging to user $i$, denoted $D \approx_i D'$.

The record-level analysis mirrors the user-level analysis, with each $f_{ij}^t$ replaced by a new $\widetilde{f}_{ij}^t$. However, sub-sampling interacts with privacy amplification by iteration, making the analysis more intricate. To characterize $\widetilde{f}_{ij}^t$, we adopt a sum-sampling operator $C_p$ defined by Definition E.1. The full proof is provided in Appendix E.

**Lemma 4.4.** *Define the trade-off function $\widetilde{f}_{ij}^{t} = T(\mathcal{A}_j^t(D), \mathcal{A}_j^t(D'))$ for $D \approx_i D'$. If each loss function is $m$-strongly convex, $M$-smooth with gradient sensitivity $\Delta$. Then, for any $\eta \in (0, 2/M)$,*

$$\widetilde{f}_{ij}^{t} \geq G\left(\frac{2\sqrt{2}c^{(t-1)K}b_K}{\eta\sigma}\right) \otimes \left[\bigotimes_{k=1}^{K} C_{\frac{b}{m_i}}\left(G\left(\frac{2a_k}{\eta\sigma}\right)\right)\right], \qquad \forall t > 1,$$

*where $m_i$ is the sample size of user $i$ and $c = \max\{|1-\eta m|, |1-\eta M|\}$. Here, the sequence $\{b_k, a_k\}$ is given by $b_{k+1} = \max\{cb_k, (1-\gamma_{k+1})(cb_k + \eta\Delta)\}$, $a_{k+1} = \gamma_{k+1}(cb_k + \eta\Delta)$ with $b_0 = 0$, and $0 < \gamma_k < 1$. If each loss function is non-convex, we have $\widetilde{f}_{ij} \geq \otimes_{k=1}^{K} C_{\frac{b}{m_i}}(G_{\frac{\Delta}{\eta\sigma}})$.*

We obtain the record-level privacy guarantee by replacing $f_{ij}^t$ in Lemma 4.1 and Theorem 4.1 with $\widetilde{f}_{ij}^t$, with details can be found in Appendix E. Moreover, our result can be extended to the non-convex case without considering privacy amplification by iteration, as detailed in Lemma E.2.

### 4.3 Secret $f$-DP with Related Noise

Our $f$-DP framework can be naturally extended to settings where users share secrets to coordinate noise addition, as proposed in Algorithm 2. We refer to this setting as Sec-$f$-LDP, formally defined in Definition 3.3. To formalize this, we consider honest-but-curious collusion at level $q$ [2], where any coalition of up to $q$ users may pool their information, including the shared secrets they have access to. This formulation models a practical threat scenario where partial trust exists across the network: users behave according to protocol but may attempt to extract additional information through collusion. We analyze privacy for Algorithm 2, which incorporates both independent Gaussian noise and pairwise correlated noise via shared secrets. The following theorem provides a quantitative guarantee under the Sec-$f$-LDP framework. The proof is given in Appendix F.2.

**Theorem 4.2.** *Algorithm 2 against honest-but-curious users colluding at level $q$ satisfies $(\mu, \mathcal{S})$-SecGDP with*

$$\mu = \Delta \cdot \sqrt{\frac{1}{(n-q)\sigma_{\text{DP}}^2} + \frac{1 - \frac{1}{n-q}}{\sigma_{\text{DP}}^2 + \lambda_2(\mathbf{L})\sigma_{\text{cor}}^2}},$$

*where, $\sigma_{\text{DP}}^2$ and $\sigma_{\text{cor}}^2$ are the variance of the independent and dependent noise correspondingly and $\mathbf{L}$ is Laplacian matrix of the graph and $\lambda_2$ is the second largest eigen-value of a matrix.*

**Remark 4.5.** In Theorem 4.2, "honest-but-curious collusion at level $q$" refers to a setting where the adversary can access all data and secrets except those of $q$ users. This follows the adversarial model adopted in [2] and is standard in secret-sharing–based privacy mechanisms. See Appendix F.1 for further details.

## 5 Experiments

In this section, we present numerical experiments to demonstrate the performance of the proposed privacy accounting methods on both synthetic and real datasets. The codes are available at `https://github.com/lx10077/PN-f-DP`. The goal is to examine how replacing previous RDP-based accounting methods with our $f$-DP-based approach affects the privacy–utility trade-off under different network settings. We first compare PN-$f$-DP with PN-RDP on synthetic graphs, then evaluate their performance on two private classification tasks, and finally analyze the improved trade-off achieved with correlated noises. Additional results on other graph structures and parameter settings are provided in Appendix H due to space limitations.

**Comparison with PN-RDP on synthetic graphs.** Following the experiment setup in [15], we generate several synthetic graphs and report the privacy parameter $\epsilon$ when we convert PN-$f$-DP or PN-RDP to $(\epsilon, \delta)$-DP for a given value of $\delta$. For fair comparison, we fix the total iterations $T$, local updates $K = 1$, and noise variance $\sigma^2 = 1$. We set $T = \Theta(\frac{\log n}{\lambda_2})$, which is the number of steps required for the random walk to converge to a minimal precision level. We use the numerical composition method in [29, 36] to convert PN-$f$-DP to $(\epsilon, \delta)$-DP. When numerically converting PN-RDP to $(\epsilon, \delta)$-DP, we adopt the method in [8], which computes the minimum $\epsilon$ among the four existing conversion methods [10, 49, 6, 4].

We report results on a hypercube graph with $n = 32$ nodes in Figure 1, and include similar results on two other synthetic graphs and one real graph in Appendix H. Figure 1(a) provides a visualization

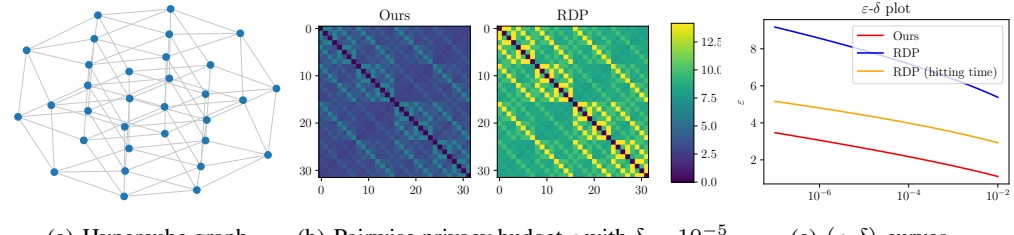

|       (a) Hypercube graph.       |  (b) Pairwise privacy budget $\epsilon$ with $\delta = 10^{-5}$.  |  (c) $(\epsilon, \delta)$-curves.  |

Figure 1: Comparisons by converting PN-$f$-DP and PN-RDP to PN-$(\epsilon, \delta)$-DP on Hypercube. See more experimental results in Appendix H.1.

of the hypercube graph. Figure 1(b) shows the pairwise privacy budgets $\epsilon$ (for $\delta = 10^{-5}$) derived by converting PN-$f$-DP and PN-RDP to $(\epsilon, \delta)$-DP. Across nearly all node pairs, the values from PN-$f$-DP are noticeably smaller—appearing darker—than those from PN-RDP, indicating stronger privacy guarantees. In Figure 1(c), we further examine the privacy leakage from user 1 to user $n$ as a function of $\delta$. In this plot, we also evaluate a refined RDP baseline that uses the exact hitting time, as opposed to the upper bound in [15]. Even with this tighter baseline, our PN-$f$-DP method consistently yields the smallest $\epsilon$, demonstrating its effectiveness in achieving tighter privacy guarantees.

**Better performance on private classification.** We then evaluate the impact of our tighter privacy accounting on downstream performance of DP-SGD through two classification tasks. The first is a **logistic regression** model trained on a binarized version of the UCI Housing dataset,[3] and the second is an **MNIST image classification** [38] task using a simple convolutional neural network. To meet a target $(\epsilon, \delta)$-DP guarantee, we compute the required gradient noise variance using different privacy accounting methods. We follow the experimental setup of [15],

| Graph | $\epsilon$ | RDP | RDP (HT) | Ours |
|---|---|---|---|---|
| Complete | 10 | 0.867 | 0.884 | **0.891** |
| Complete | 8 | 0.854 | 0.876 | **0.885** |
| Complete | 5 | 0.813 | 0.852 | **0.869** |
| Expander | 10 | 0.797 | 0.823 | **0.854** |
| Expander | 8 | 0.763 | 0.797 | **0.840** |
| Expander | 5 | 0.647 | 0.706 | **0.792** |

Table 1: Test accuracy on MNIST classification with $\delta = 10^{-5}$. Bold denotes the best.

including standardizing feature vectors, normalizing data points, and splitting each dataset randomly into 80% training and 20% test sets. The training data is then distributed across $n = 2^8$ users, each holding 64 local samples, according to an expander graph topology. We compare three approaches: decentralized DP-GD, local DP-GD, and our random-walk-based DP-GD under different noise variances. We focus on the privacy loss from user 1 to user 2, fixing $(\epsilon, \delta) = (10, 10^{-5})$, and record both objective values and test accuracy every 100 iterations. Results are presented in Figure 2. As seen, the random-walk-based method with $f$-DP consistently achieves the best performance. This improvement is due to the tighter noise calibration enabled by $f$-DP, which allows us to meet the same $(\epsilon, \delta)$-DP guarantee with lower noise variance—thus enhancing the privacy-utility trade-off. Table 1 presents the test accuracy results for the MNIST classification task under varying graph structures and privacy budgets $\epsilon$. As shown, our method consistently yields higher accuracy across settings. The same pattern holds for logistic regression when increasing the number of users to $n = 2^{11}$ or reducing the privacy budget to $\epsilon = 5$ or 3. These additional results are reported in Appendix H.2.

**Better performance on related noises.** Finally, we demonstrate how our $f$-DP method improves the privacy-utility trade-off for decentralized FL with related noises. Our experimental setup follows [2]. We consider $n = 16$ users on two standard network topologies with increasing connectivity: a ring and a 2D torus (grid). Each experiment is repeated with four random seeds for reproducibility. We compare against two baselines: SecLDP [2] and local DP. Figure 3 shows the results for logistic regression on the a9a dataset [11]. As seen, our method (red curves) consistently achieves a better privacy-utility trade-off across different values of the privacy budget $\epsilon$ and both network topologies.

## 6 Conclusions

We introduced two new differential privacy notions for decentralized federated learning: PN-$f$-DP and Sec-$f$-LDP, extending the $f$-DP framework to settings with independent and correlated noise, respectively. These enable tighter and composable privacy analysis for various decentralized protocols.

---

[3]Available at `https://www.openml.org/search?type=data&sort=runs&id=823/`.

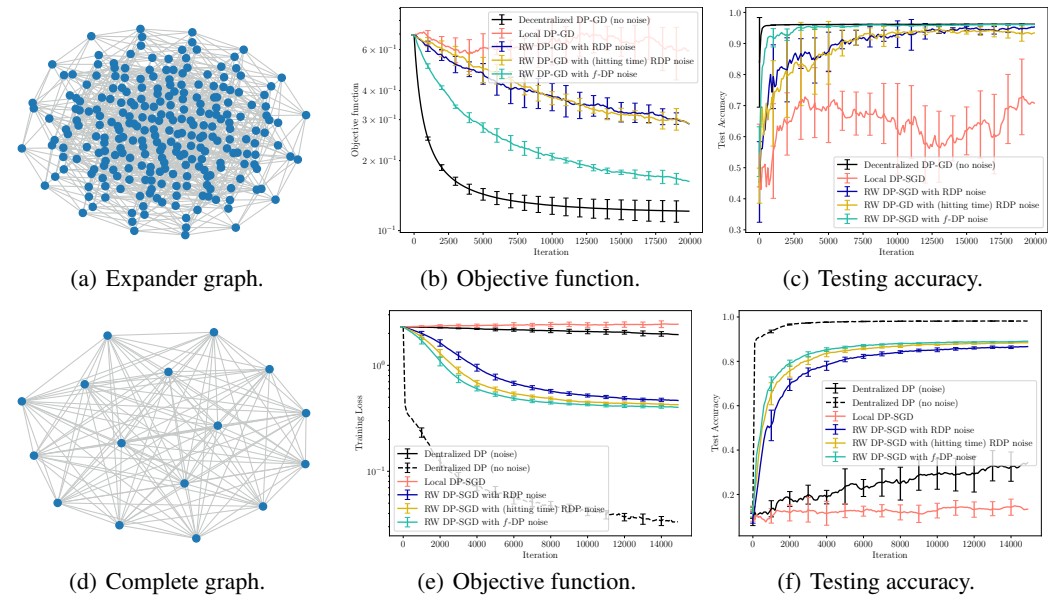

(a) Expander graph.  (b) Objective function.  (c) Testing accuracy.

(d) Complete graph.  (e) Objective function.  (f) Testing accuracy.

Figure 2: **Top:** Performance of private logistic regression on the Housing dataset. **Bottom:** Performance of private MNIST classification using a neural network.

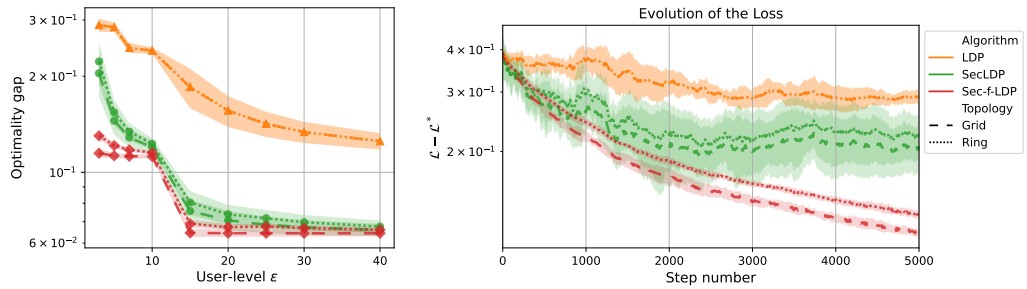

Figure 3: **Left**: Privacy-utility trade-offs for our $f$-DP, DECOR, and LDP when converted to the same $(\epsilon, 10^{-5})$-DP. **Right**: Optimality gap vs. iteration steps for Alg. 2 using different privacy accounting methods with $\epsilon = 3$. Similar results for other values of $\epsilon \in \{5, 7, 10\}$ are provided in Appendix H.3.

Using PN-$f$-DP, we derived user- and record-level guarantees for decentralized DP-SGD with random-walk communication, while Sec-$f$-LDP was applied to correlated-noise algorithms such as DecoR. Experiments on synthetic graphs and the UCI housing dataset show that $f$-DP–based accounting consistently achieves a stronger privacy–utility trade-off than RDP-based methods. Our framework readily generalizes to dynamic or time-varying graphs by updating the transition matrices $W^t$ each round and applies to other decentralized FL algorithms—such as gossip-based, asynchronous, or push-sum protocols—by defining suitable $W^t$ and update rules. It also accommodates non-uniform or partial participation, and known participation probabilities can be incorporated into the weighting scheme for additional privacy amplification. Extending these ideas and establishing information-theoretic lower bounds remain promising directions for future work.

## Acknowledgments

This work was supported in part by NIH grants U01CA274576 and R01EB036016, NSF grant DMS-2310679, a Meta Faculty Research Award, and Wharton AI for Business. The content is solely the responsibility of the authors and does not necessarily represent the official views of the NIH.

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

# A  Preliminaries on Markov Chains

Here, we summarize several properties of Markov chains that will be utilized in our proof, as detailed in standard Markov chain textbooks, such as [39].

**Definition A.1** (Irreducible). A Markov chain with transition matrix $W$ is called irreducible if there exists $t > 0$ such that $W_{ij}^t > 0$ for any $i, j \in [n]$.

For a probability distribution $\pi$ supported on $[n]$ with pmf $\{\pi_i\}_{i=1}^n$ (i.e., $\pi_i \geq 0$ and $\sum_{i=1}^n \pi_i = 1$) and a transition matrix $W$, we define $\pi(W)$ as

$$(\pi(W))_j = \sum_{i=1}^n \pi_i W_{ij}, \qquad \text{for any } j \in [n].$$

**Definition A.2** (Stationary distribution). For a Markov chain with transition matrix $W$, we say $\pi$ is a stationary distribution of $W$ is $\pi(W) = W$.

**Proposition A.1** (Existence of a stationary distribution). *For an irreducible Markov chain with transition matrix $W$, there is a probability $\pi$ such that $\pi(W) = W$, i.e., $\pi$ is a stationary distribution of $W$.*

Let $\mathcal{T}(i) = \{t \geq 1 : (W^t)_{ii} > 0\}$ be the set of all times when the probability of the chain starting from node $i$ returns to node $i$. The period of node $i$ is defined as the greatest common divisor of $\mathcal{T}(i)$.

**Definition A.3.** We say a Markov chain is aperiodic if the period of every node $i$ is 1.

# B  Useful Properties of $f$-DP

## B.1  Joint Concavity of Trade-Off Functions

As the random walk implies a mixture distribution, we first provide $f$-DP guarantees for mixture mechanisms using the joint concavity of trade-off functions introduced by [61].

Let $\{P_i\}_{i=1}^m$ and $\{Q_i\}_{i=1}^m$ be two sequences of probability distributions and let $p_i$ and $q_i$ be the pdf of $P_i$ and $Q_i$, correspondingly. A mixture distribution $P_{\mathbf{w}}$ that is a mixture of $\{P_i\}_{i=1}^m$ with weights $\mathbf{w} = \{w_i\}_{i=1}^m$ has pdf $p_{\mathbf{w}} = \sum_{i=1}^m w_i p_i$. Similarly, let $Q_{\mathbf{w}}$ denote the mixture of $\{Q_i\}_{i=1}^m$ with the same weights $\mathbf{w} = \{w_i\}_{i=1}^m$, i.e., $Q_{\mathbf{w}}$ has a pdf given by $q_{\mathbf{w}} = \sum_{i=1}^m w_i q_i$.

**Lemma B.1** (Joint concavity of trade-off functions[61]). *For two mixture distributions $P_{\mathbf{w}}$ and $Q_{\mathbf{w}}$, it holds*

$$T(P_{\mathbf{w}}, Q_{\mathbf{w}})(\alpha(t, c)) \geq \sum_{i=1}^m w_i T(P_i, Q_i)(\alpha_i(t, c)),$$

*where $\alpha_i(t, c) = \mathbb{P}_{X \sim P_i}\left[\frac{q_i}{p_i}(X) > t\right] + c\mathbb{P}_{X \sim P_i}\left[\frac{q_i}{p_i}(X) = t\right]$ is the type I error for testing $P_i$ v.s. $Q_i$ using the likelihood ratio test and $\alpha(t, c) = \sum_{i=1}^m w_i \alpha_i(t, c)$.*

According to [61], Lemma B.1 implies the joint convexity of $F$-divergences, including the scaled exponentiation of the Rényi divergence and the hockey-stick divergence.

## B.2  Privacy Amplification by Iteration

Consider the DP-SGD algorithm defined iteratively as

$$\theta_{k+1} = \Pi_{\mathcal{K}}\left[\theta_k - \eta\left(g_k(\theta_k) + \mathcal{N}(0, \sigma^2)\right)\right]$$

with some initialization $\theta_0$, some closed and convex set $\mathcal{K}$, and some gradient map $g_k$. The effect of iteration on the Gaussian noise has been studied by [26], which is termed as privacy amplification by iteration. This is further investigated by [3, 68] which shows that the privacy budget of DP-SGD may converge as the iteration goes. Here we introduce recent $f$-DP result for privacy amplification by iteration [8] which will be adopted in our privacy analysis.

For a map $\phi$, we say $\phi$ is a contraction $\phi$ is $c$-Lipschitz for some $0 < c < 1$. When $c = 1$, we say $\phi$ is non-expansive. The following conclusion is a well-known result to guarantee that the gradient map $\theta \mapsto \theta + g_k(\theta)$ is non-expansive.

**Lemma B.2.** *Consider a loss function $\ell$ that is convex and $M$-smooth. Then, the gradient descent update $\phi(x) = x - \eta\nabla f(x)$ is non-expansive for each $\eta \in [0, 2/M]$. If $f$ is additionally $m$-strongly convex and $\eta \in (0, 2/M)$, then $\phi$ is $c$-Lipschitz with $c = \max\{|1 - \eta m|, |1 - \eta M|\} < 1$.*

The iteration of DP-SGD relates to the contractive noisy iteration.

**Definition B.1** (Contractive noisy iterations (CNI))**.** The contractive noisy iterations $\text{CNI}(\theta_0, \{\phi_k\}_{k\in[t]}, \{\xi_k\}_{k\in[t]}, \mathcal{K})$ corresponding to a sequence of contractive functions $\{\phi_k\}_{k\in[t]}$, a sequence of noise distributions $\{\xi_k\}_{k\in[t]}$, and a closed and convex set $\mathcal{K}$, is the stochastic process

$$\theta_{k+1} = \Pi_{\mathcal{K}}(\phi_{k+1}(\theta_k) + Z_{k+1}), \tag{1}$$

where $Z_{k+1} \sim \xi_{k+1}$ is independent of $(\theta_0, \ldots, \theta_k)$.

For CNI, one can consider the privacy amplification by iteration. Here, we apply the $f$-DP guarantee in the following lemma from [8].

**Lemma B.3** (Privacy amplification by iteration, Lemma C.3 in [8])**.** *Let $\theta_t$ and $\theta'_t$ respectively be the output of $\text{CNI}(\theta_0, \{\phi_k\}_{k\in[t]}, \{\mathcal{N}(0, \sigma^2\mathbf{I}_p)\}_{k\in[t]}, \mathcal{K})$ and $\text{CNI}(\theta_0, \{\phi'_k\}_{k\in[t]}, \{\mathcal{N}(0, \sigma^2\mathbf{I}_p)\}_{k\in[t]}, \mathcal{K})$ such that each $\phi_k, \phi'_k$ is $c-$Lipschitz and $\|\phi_k - \phi'_k\|_\infty \le s_k$ for all $k \in [t]$. Then for any intermediate time $\tau$ and shift parameters $\gamma_{\tau+1}, \cdots, \gamma_t \in [0, 1]$ with $\gamma_t = 1$,*

$$T(\theta_t, \theta'_t) \ge G\left(\frac{1}{\sigma}\sqrt{\sum_{k=\tau+1}^{t} a_k^2}\right),$$

*where $a_{k+1} = \gamma_{k+1}(cb_k + s_{k+1})$, $b_{k+1} = (1 - \gamma_{k+1})(cb_k + s_{k+1})$, and $\|\theta_\tau - \theta'_\tau\| \le b_\tau$.*

### B.3 Additional Useful Properties

**Post-processing property of $f$-DP [17].** An important property of DP is its resistance to data-independent post-processing. For $f$-DP, this post-processing property is described in the following proposition.

**Proposition B.1.** *For any probability distributions $P, Q$ and any data-independent post-processing Proc, it holds*

$$T(Proc(P), Proc(Q)) \ge T(P, Q).$$

**From $f$-DP to $(\epsilon, \delta)$-DP and back [17].** On one hand, a $f$-DP mechanism with a symmetric trade-off function $f$ is $(\epsilon, \delta(\epsilon))$-DP for all $\epsilon > 0$ with

$$\delta(\epsilon) = 1 + f^*(-e^\epsilon).$$

On the other hand, if a mechanisms is $(\epsilon, \delta(\epsilon))$-DP, then it is $f$-DP with $f = \sup_{\epsilon>0} f_{\epsilon,\delta}$ where

$$f_{\epsilon,\delta}(\alpha) = \max\left\{0, 1 - \delta - e^\epsilon\alpha, e^{-\epsilon(1-\delta-\alpha)}\right\}.$$

**From $f$-DP to RDP [17].** An $f$-DP mechanisms is $(\alpha, \epsilon_f(\alpha))$-RDP where

$$\epsilon_f(\alpha)) = \frac{1}{\alpha - 1}\log\int_0^1 |f'(x)|^{1-\alpha}dx.$$

In particular, a $\mu$-GDP mechanism is $(\alpha, \frac{1}{2}\mu^2\alpha)$-RDP. It is worth mentioning that converting RDP back to $f$-DP is challenging.

## C  Additional Details on the Extension to Record-Level Privacy

## D  Proofs of Section 4.1

The proof of the user-level analysis can be divided into the following steps: first, we prove Lemma 4.1, which deals with the mixture distributions caused by random walk; then, we study the privacy amplification of Algorithm 1 and prove Lemma 4.2; the last step is to prove Theorem 4.1.

## D.1 Proof of Lemma 4.1

Recall the random variable $\mathcal{A}_j^{\text{single}}(D)$ that represents the model visits user $j$ for the first time after passing user $i$ and $\mathcal{A}_j^t(D)$ which represents the model visits user $j$ for the first time at the $t$-th iteration. Thus, one has

$$\mathbb{P}[\mathcal{A}_j^{\text{single}}(D) = \mathcal{A}_j^t(D)] = \mathbb{P}[\tau_{ij} = t] = w_{ij}^t, \qquad \text{for all } 1 \le t \le T.$$

where $\tau_{ij}$ is the hitting time from $i$ to $j$. There is a chance that the model will not be observed after $T$ steps, which is represented by a random variable $\mathcal{A}_j^{T+1}(D)$. And we have

$$\mathbb{P}[\mathcal{A}_j^{\text{single}}(D) = \mathcal{A}_j^{T+1}(D)] = \mathbb{P}[\tau_{ij} > t] = 1 - \sum_{t=1}^{T} w_{ij}^t =: w_{ij}^{T+1}, \qquad \text{for all } 1 \le t \le T.$$

By the definition of mixture distributions, one has $\mathcal{A}_j^{\text{single}}(D)$ is a mixture of $\{\mathcal{A}_j^t(D)\}_{t=1}^{T+1}$ with weights $\{w_{ij}^{t+1}\}_{t=1}^{T+1}$. We finish the proof by applying the joint convexity in Lemma B.1 to the mixture above.

## D.2 Proof of Lemma 4.2

Recall that $f_{ij}^t$ is the privacy loss when the model is observed in the user $j$ after $t$ steps. Note that, after $t$ steps, Algorithm 1 has been run for $tK$ iterations. Thus, the trade-off function $f_{ij}^t$ can be bounded by adopting privacy amplification by iteration in [8]. Using Lemma B.3, we have the following iteration results for Algorithm 1.

**Lemma D.1** (Privacy amplification for decentralized DP-SGD). *Under assumptions in Lemma 4.2, for any $\gamma_1, \cdots, \gamma_{tK} \in [0, 1]$ with $\gamma_{tK} = 1$, it holds*

$$f_{ij}^{tK} \ge G_{\frac{1}{\eta\sigma}\sqrt{\sum_{k=1}^{tK} a_k^2}},$$

*where $a_{k+1} = \gamma_{k+1}(cb_k + s_{k+1})$, $b_{k+1} = (1 - \gamma_{k+1})(cb_k + s_{k+1})$ with $b_0 = 0$, $s_k \equiv \eta\Delta$ for $1 \le k \le K$, and $s_k \equiv 0$ for $K + 1 \le k \le Kt$. Here $G_{\frac{1}{\eta\sigma}\sqrt{\sum_{k=1}^{tK} a_k^2}}$ is the Gaussian trade-off function $G_\mu$ with $\mu = \frac{1}{\eta\sigma}\sqrt{\sum_{k=1}^{tK} a_k^2}$.*

*Proof.* The proof is based on Lemma B.3. To obtain the final result, we need to clarify the choice of parameters $s_k$ and $b_\tau$ in Lemma B.3 with $\tau = 0$. Recall that $s_k$ is the upper bound on $\|\phi_k - \phi_k'\|_\infty$, where $\phi_k(x) := x - \eta/b \sum_{i=1}^{b} \nabla f_i(x)$ represents the gradient map and $\phi_k' := x - \eta/b \sum_{i=1}^{b} \nabla f_i'(x)$ is another gradient map obtained by modifying the dataset held by user $i$. In the first $K$ steps ($1 \le k \le K$) the iteration is running with user $i$'s dataset. Thus, $s_k$ is upper bounded $\eta\Delta$ where $\Delta$ is the sensitivity of the sub-sampled gradient.

After $K$ iterations, for $K + 1 \le k \le tK$, the model has been sent to other users and changing the data records in user $i$ will not change the gradient. Thus, $s_k$ becomes 0. There is a possibility that the model might be sent back to user $i$ again for $2K + 1 \le k \le tK$ before it reaches user $j$. However, the privacy loss of the second pass is accounted for using other $f_{ij}^{\text{single}}$ functions within the total number $\lceil T/n + \zeta \rceil$ of trade-off functions $f_{ij}^{\text{single}}$s and will be considered using composition in Theorem 4.1 . Therefore, multiple passes through user $i$ will not affect the privacy of the current single pass and we still have $s_k = 0$ even though it might be sent back to user $i$ again for $2K + 1 \le k \le tK$. We finish the proof by replacing $\sigma$ in Lemma B.3 with $\eta\sigma$ as there is a scalor $\eta$ of the Gaussian noise in Algorithm 1. $\qquad\square$

The finish the proof of Lemma 4.2, motivated by [8], is enough to minimize the term $\sum_{k=1}^{tK} a_k^2$ in Lemma D.1, which is given in the following lemma.

**Lemma D.2.** *Given $0 < c < 1$, the optimal value of*

$$\min \sum_{k=1}^{tK} a_k^2$$

*subject to*

$$a_{k+1} = \gamma_{k+1}(cb_k + s_{k+1}) \geq 0, \quad b_{k+1} = (1 - \gamma_{k+1})(cb_k + s_{k+1}) \geq 0, \quad b_0 = 0, \quad b_{tK} = 0,$$
$$0 \leq \gamma_k \leq 1, \quad s_k \equiv s, \text{ for } 0 \leq k \leq K, \quad \text{and } s_k = 0, \text{ for } K + 1 \leq k \leq tK,$$

*is* $c^{2K(t-1)} \frac{1+c}{1-c} \cdot \frac{(1-c^K)^2}{1-c^{2tK}} s^2$.

*Proof.* Note that

$$b_{tK} + \sum_{k=1}^{tK} c^{tK-k} a_k = b_0 + \sum_{k=1}^{tK} c^{tK-k} s_k, \qquad \text{and}$$

$$\sum_{k=1}^{tK} c^{tK-k} a_k = c^{K(t-1)} \cdot \frac{1-c^K}{1-c} s.$$

By the Cauchy-Schwarz inequality, we have

$$\sum_{k=1}^{tK} a_k^2 \geq \frac{\left(\sum_{k=1}^{tK} c^{tK-k} a_k\right)^2}{\sum_{k=1}^{tK} c^{2(tK-k)}} = c^{2K(t-1)} \cdot \frac{1+c}{1-c} \cdot \frac{(1-c^K)^2}{1-c^{2tK}} s^2,$$

where the equality holds when $\gamma_k = 1$, $a_k = s$ for $0 \leq k \leq K$ and $\gamma_k = 0$, $a_k = 0$ for $K + 1 \leq k \leq Kt$. $\qquad\square$

We finish the proof of Theorem 4.1 by taking $s = \eta\Delta$ in Lemma D.2 and combining Lemma D.2 with Lemma D.1.

### D.3 Proof of Lemma 4.3

**Lemma D.3.** *For two random variables $X, X'$ satisfying $T(X, X') \geq T(\zeta, \zeta')$ with $\zeta \sim \mathcal{N}(0, \sigma_1^2), \zeta' \sim \mathcal{N}(\mu, \sigma_1^2)$ for some $\mu > 0$, we have $T(X + \xi, X' + \xi') \geq T(\zeta + \xi, \zeta' + \xi')$ for $\xi, \xi' \sim \mathcal{N}(0, \sigma_2^2)$.*

*Proof.* By Theorem 2.10 in [17], we have $X = proc(\zeta)$ and $X' = proc(\zeta')$ for some post-processing map $proc$. The proof is done by noting that there is a new post-processing new processing $proc'$ : $\zeta + \xi \mapsto proc(\zeta) + \xi$ such that $X + \xi = proc'(\zeta + \xi)$. $\qquad\square$

*Proof of Lemma 4.3.* After $K$ iterations in user $i$, we have $f_{ij}^0 \geq G_{\mu_{i,K}}$ with $\mu_{i,K} = \sqrt{K}\Delta/\sigma$. Rewrite $G_{\mu_{i,K}} = T(\mathcal{N}(0, \sigma^2), \mathcal{N}(\sigma\mu_{i,K}, \sigma^2))$. By Lemma D.3, we have $f_{ij}^1 \geq T(\mathcal{N}(0, 2\sigma^2), \mathcal{N}(\sigma\mu_{i,K}, 2\sigma^2)) = G_{\mu_1}$ with $\mu_1 = \mu_{i,K}/\sqrt{2}$. If $f_{ij}^{t-1} \geq G_{\mu_{tK-1}}$ with $\mu_{tK-1} = \mu_{i,K}/\sqrt{tK}$. Then, we can write $G_{\mu_{tK-1}} = T(\mathcal{N}(0, \sqrt{t}\sigma^2), \mathcal{N}(\mu_{i,K}\sigma, \sqrt{t}\sigma^2))$. Thus, by Lemma D.3, we obtain $f_{ij}^t \geq T(\mathcal{N}(0, (tK + 1)\sigma^2), \mathcal{N}(\mu_{i,K}\sigma, (tK + 1)\sigma^2)) = G_{\mu_t}$. We finish the proof by induction. $\qquad\square$

### D.4 Proof of Theorem 4.1

**Lemma D.4** (Hoeffding's inequality for Markov chain, Theorem 1 in [23])**.** *Let $\{Y_t\}_{t \geq 0}$ be a Markov chain with transition matrix $W$ and stationary distribution $\pi$. For any $T > 0$ and a sequence of bounded functions $\{f_t\}_{t=1}^T$ such that $\sup_x |f_t(x)| \leq M$, it holds*

$$\mathbb{P}_\pi \left[ \sum_{t=1}^T (f_t(Y_t) - \mathbb{E}_{Y \sim \pi}(f_t(Y))) \geq \zeta \right] \leq \exp\left( -\frac{1-\lambda_2}{1+\lambda_2} \frac{2\zeta^2}{TM^2} \right),$$

*for any $\zeta > 0$, where $\lambda_2$ is the second largest eigen-value of $W$.*

*Proof of Theorem 4.1.* Let $f_t$ be an indicator function that equals 1 if $X_t$ visits $j$ at time $t$, and 0 otherwise. Then $\sum_{t=1}^{T}(f_t(Y_t))$ is the total number of visits to the user $j$, which means the model is observed at user $j$ for $\sum_{t=1}^{T}(f_t(Y_t))$ times and the total privacy loss is $\sum_{t=1}^{T}(f_t(Y_t))$-fold composition of $f_{ij}^{\text{single}}$. Under assumptions of Theorem 4.1, the stationary distribution $\pi$ is uniformly distributed on $[n]$ (cf., Exercise 1.7 in [39]). Thus, we have $\mathbb{E}_{Y \sim \pi}(f_t(Y)) = 1/n$ in Lemma D.4. Note that $\|f_t\|_\infty \leq M = 1$. As a result of Lemma D.4, one has $\sum_{t=1}^{T}(f_t(Y_t)) \geq (1+\zeta)T/n$ with probability $\exp\left(-\frac{1-\lambda_2}{1+\lambda_2}\frac{2\zeta^2 T}{n^2}\right)$ and we finish the proof. $\qquad\square$

**Remark D.1.** When converting $f$-DP to $(\epsilon, \delta)$-DP, $\delta'_{T,n}$ contributes an additional term to $\delta$. The proof employs Hoeffding's inequality for general Markov chains [23], which achieves faster exponential decay of $\delta'_{T,n}$ by exploiting the spectral gap, improving upon the polynomial decay in the classical result (Theorem 12.21 in [39]).

# E    Proofs of Section 4.2

The formal statement of Theorem E.1 is given below.

**Definition E.1** ($p$-sampling operator of $f$-DP [17])**.** For a trade-off function $f$ and a sampling parameter $0 < p < 1$, we let $f_p = pf + (1-p)\text{Id}$. The $p$-sampling operator $C_p$ that maps a trade-off function to another trade-off function is defined as $C_p(f) = (\min\{f_p, f_p^{-1}\})^{**}$, where $(\cdot)^{-1}$ is the left inverse of a non-increasing function and $(\cdot)^{**}$ is the double convex conjugate of a function.

**Theorem E.1.** *Assume the conditions of Theorem 4.1 hold. For all $s \in [0, \infty)$, let*

$$\widetilde{f}_{ij}^{\text{single}}(\alpha(s)) := \sum_{t=1}^{T} w_{ij}^{t}\widetilde{f}_{ij}^{t}(\alpha_t(s)) + w_{ij}^{T+1}(1 - \alpha_{T+1}(s)),$$

*where $\alpha_t(s)$ and $\alpha(s)$ are the same defined in Lemma 4.1. Then, for $D \approx_i D'$, we have* $T(\mathcal{A}_j(D), \mathcal{A}_j(D')) \geq \left(\widetilde{f}_{ij}^{\text{single}}\right)^{\otimes \lceil (1+\zeta)T/n \rceil}$ *with probability $1 - \delta'_{T,n}$, for any $\zeta > 0$.*

The proof of record-level privacy in Theorem E.1 necessitates iteration analysis with subsampling. The proof closely follows the approach in [8]. However, in federated learning, subsampling affects only the first $K$ iterations, as the subsequent $(t-1)K$ iterations do not use the dataset from user $i$, which differs from [8]. Due to subsampling, the Gaussian mechanisms might be influenced by a Bernoulli distribution. Therefore, we begin by introducing the following lemma from [8].

**Lemma E.1** (Lemma C.12 in [8])**.** *For $s \geq 0$ and $0 \leq p \leq 1$, let*

$$R(s, \sigma, p) = \inf\{T(VW + Z, VW' + Z) : V \sim \text{Bern(p)}, \|\mathrm{W}\|, \|\mathrm{W}'\| \leq \mathrm{s}, Z \sim \mathcal{N}(0, \sigma^2 \mathrm{I_d})\},$$

*where the infimum is taken pointwisely and is over independent $V, W, W', Z$. Then, one has $R(s, \sigma, p) \geq C_p(G(\frac{2s}{\sigma}))$.*

The proof of Lemma 4.4 requires the following proposition. We adopt the notation from [8] and divide the minibatch subsample into two subsets, $R_k$ and $C_k$, as follows. For $x^*$, a data record that might be changed by an attacker under the setting of record-level DP, we use the following definition of $R_k$ and $C_k$ from [8]:

1. Sample a set $A_1$ of size $b$ in $D_i \setminus \{x^*\}$ uniformly at random.

2. Sample an element $A_2$ from $A_1$ uniformly at random. This element will serve as a candidate to be (potentially) replaced by $x^*$.

3. Let $R_k = A_1 \setminus \{A_2\}, C_k = A_2$.

Following [8], we also define the following stochastic version of the CNI. In the $k$-th iteration, $D$ and $D'$ differ by a single data record held by the user $i$. Denote the record as $C_k$ when used in the $k$-th iteration, and as $C'_k$ when not used in the mini-batch of the $k$-th iteration. Let the index of the mini-batch of size $b$ be $S_k = (R_k, C_k) \in [m_i]$ or $S'_k = (R_k, C'_k)$. Let $\ell_z$ be the loss function

corresponding to a single datapoint $z$. Denote

$$\phi_{S_k}(\theta) = \theta - \frac{\eta}{b}(\nabla \ell_{C_k} + \sum_{z \in R_k} \nabla \ell_z)(\theta),$$

$$\phi'_{S_k}(\theta) = \theta - \frac{\eta}{b}(\nabla \ell'_{C_k} + \sum_{z \in R_k} \nabla \ell_z)(\theta),$$

$$\psi_{S_k}(\theta) = \theta - \frac{\eta}{b} \sum_{i \in R_k \cup C'_k} \nabla \ell_z(\theta).$$

**Proposition E.1.** *Assume that the loss functions are $m$-strongly convex, $M$-smooth loss gradient sensitivity $\Delta$. Then, for any $\eta \in (0, 2/M)$, $t > 1$, Algorithm 1 is $f$-DP, where*

$$f = G\left(\frac{2\sqrt{2}cb_{tK-1}}{\eta\sigma}\right) \otimes \bigotimes_{k=1}^{tK-1} C_{p_k}\left(G\left(\frac{2a_k}{\eta\sigma}\right)\right)$$

*for any sequence $\{b_k, a_k, s_k\}$ given by*

$$b_{k+1} = \max\{cb_k, (1 - \gamma_{k+1})(cb_k + s_k)\},$$
$$a_{k+1} = \gamma_{k+1}(cb_k + s_k),$$
$$s_k = \max\{\|\phi_{S_k} - \psi_{S_k}\|_\infty, \|\phi'_{S'_k} - \psi_{S'_k}\|_\infty\},$$

*and $b_0 = 0$, $0 \le \gamma_k \le 1$, $c = \max\{|1 - \eta m|, |1 - \eta M|\}$, $s_k = 0$, $p_k = b/m_i$ for $1 \le k \le K$ and $p_k = 0$ for any $k > K$.*

*Proof.* The iterates of Noisy SGD with respect to $\{\ell_z\}_{z \in D}$ and $\{\ell'_z\}_{z \in D'}$ are given by

$$\theta_{k+1} = \Pi_\mathcal{K}(\psi_{S_k}(\theta_k) + V_k(\phi_{S_k} - \psi_{S_k})(\theta_k) + Z_{k+1}),$$
$$\theta'_{k+1} = \Pi_\mathcal{K}(\psi_{S'_k}(\theta'_k) + V'_k(\phi'_{S'_k} - \psi_{S'_k})(\theta'_k) + Z'_{k+1}),$$

where $Z_{k+1}, Z'_{k+1} \sim \mathcal{N}(0, \eta^2\sigma^2 I_d)$ and $V_k, V'_K \sim \text{Bern}(p_k)$. For any $0 \le \gamma_k \le 1$, we consider shifted interpolated processes introduced by [8] which is defined as

$$\widetilde{\theta}_{k+1} = \Pi_\mathcal{K}(\psi_{S_k}(\widetilde{\theta}_k) + \gamma_{k+1}V_k(\phi_{S_k}(\theta_k) - \psi_{S_k}(\widetilde{\theta}_k)) + Z_{k+1}),$$
$$\widetilde{\theta}'_{k+1} = \Pi_\mathcal{K}(\psi_{S'_k}(\widetilde{\theta}'_k) + \gamma_{k+1}V'_k(\phi'_{S'_k}(\theta'_k) - \psi_{S'_k}(\widetilde{\theta}'_k)) + Z'_{k+1}),$$

with $\widetilde{\theta}_0 = \widetilde{\theta}'_0 = \theta_0$. Using Lemma C.14 in [8], we have the trade-off function between $\widetilde{\theta}_{tK-1}$ and $\widetilde{\theta}'_{tK-1}$ is given by

$$T(\widetilde{\theta}_{tK-1}, \widetilde{\theta}'_{tK-1}) \ge \bigotimes_{k=1}^{tK-1} C_{p_k}\left(G\left(\frac{2a_k}{\eta\sigma}\right)\right)$$

for any sequence $\{b_k, a_k, s_k\}$

$$b_{k+1} = \max\{cb_k, (1 - \gamma_{k+1})(cb_k + s_k)\},$$
$$a_{k+1} = \gamma_{k+1}(cb_k + s_k), \tag{2}$$
$$s_k = \max\{\|\phi_{S_k} - \psi_{S_k}\|_\infty, \|\phi'_{S'_k} - \psi_{S'_k}\|_\infty\},$$

with $b_0 = 0$ and $0 \le \gamma_k \le 1$. By the ralationship between the original iteration and the interpolation (Section C.3 in [8]), one has

$$T(\theta_{tK}, \theta'_{tK}) \ge T(\widetilde{\theta}_{tK-1}, \widetilde{\theta}'_{tK-1}) \otimes G\left(\frac{2\sqrt{2}cb_{tK-1}}{\eta\sigma}\right).$$

This completes the proof of Proposition E.1. $\qquad\square$

*Proof of Lemma 4.4.* Similar to the proof of Lemma D.1, for Algorithm 1, we have $s_k = 0$ for any $K + 1 \leq k \leq Kt$ and $s_k = \eta\Delta$ for any $k \leq K$. In the first $K$ steps ($1 \leq k \leq K$) the iteration is running with two datasets, $D_i$ and $D_i'$, which differ by a a single data record belonging to user $i$. Thus, $p_k = b/m_i$ for $k \leq K$ is the probability of selecting different data in the mini-batch of size $b$ from all $m_i$ data belonging to user $i$. After $K$ iterations, for $K + 1 \leq k \leq tK$, the model has been sent to other users and changing a single data record in user $i$ will not change the gradient. Thus, $p_k$ becomes 0.

Therefore, Lemma 4.4 follows directly by letting $s_k = 0$, $p_k = 0$ for any $k > K$ and $s_k = \eta\Delta$, $p_k = b/m_i$ for any $k \leq K$. Overall, we obtain

$$f = G\left(\frac{2\sqrt{2}cb_{tK-1}}{\eta\sigma}\right) \otimes \bigotimes_{k=1}^{tK-1} C_{p_k}\left(G\left(\frac{2a_k}{\eta\sigma}\right)\right)$$

$$= G\left(\frac{2\sqrt{2}cb_{tK-1}}{\eta\sigma}\right) \otimes \bigotimes_{k=1}^{K} C_{p_k}\left(G\left(\frac{2a_k}{\eta\sigma}\right)\right) \otimes \bigotimes_{k=K+1}^{Kt-1} G\left(\frac{2}{\eta\sigma}\sqrt{\sum_{k=K}^{tK-1} a_k^2}\right)$$

$$= G\left(\frac{2\sqrt{2}c^{(t-1)K}b_K}{\eta\sigma}\right) \otimes \bigotimes_{k=1}^{K} C_{p_k}\left(G\left(\frac{2a_k}{\eta\sigma}\right)\right).$$

This completes the proof of Lemma 4.4. □

**Lemma E.2.** *For non-convex loss functions, we have*

$$\widetilde{f}_{ij}^t \geq T(\mathcal{A}_j^0(D) + \mathcal{N}(0, Kt\sigma^2), \mathcal{A}_j^0(D') + \mathcal{N}(0, Kt\sigma^2)).$$

*Proof of Lemma E.2.* After $K$ iterations in user $i$, we have $\widetilde{f}_{ij}^0 = T(\mathcal{A}_j^0(D), \mathcal{A}_j^0(D')) \geq \otimes_{k=1}^K C_{b/m_i}(G_{\Delta/\eta\sigma})$. According to Lemma D.3, we have

$$\widetilde{f}_{ij}^t \geq T(\mathcal{A}_j^0(D) + \mathcal{N}(0, Kt\sigma^2), \mathcal{A}_j^0(D') + \mathcal{N}(0, Kt\sigma^2)).$$

□

# F Details of Secrect-based DP

## F.1 Details of Secret-Based Local Differential Privacy

In the framework of secret-based differential privacy (DP) [2], the objective is to safeguard user data privacy against an adversary capable of eavesdropping on all communications. Each connected pair of users $i, j \in \mathcal{V}$ shares a sequence of secrets $S_{ij}$, which are realizations of random variables known exclusively to the two corresponding nodes. In practice, these secrets can be locally generated from shared random seeds exchanged during an initial round of encrypted communication [9]. Conceptually, one may view the secrets themselves as these shared randomness seeds. We denote by $\mathcal{S} := \{S_{ij} : i, j \in \mathcal{V}\}$ the collection of all such secrets. For a subset of nodes $\mathcal{I} \subset \mathcal{V}$, we denote by $\mathcal{S}_\mathcal{I} = \{S_{ij}, i, j \in \mathcal{V}, i, j \notin \mathcal{I}\}$ the set of secrets hidden from all users in $\mathcal{I}$. Based on $\mathcal{S}_\mathcal{I}$, we can formally introduce Definition 3.3. We consider an honest-but-curious setting, where any subset of users of size $q < n$ may collude by sharing all the secrets to which they have access.

**Definition F.1** (Sec-$f$-LDP against honest-but-curious users colluding at level $q$.)**.** We say an algorithm $\mathcal{A}$ satisfy Sec-$f$-LDP against honest-but-curious users colluding at level $q$ if it satisfies $\mathcal{S}_\mathcal{I}$-Sec-$f$-LDP for any $\mathcal{I}$ with $|\mathcal{I}| \leq q$.

## F.2 Proof of Section 4.3

*Proof of Theorem 4.2.* Suppose there are $q$ out of $n$ users that are colluding. We assume that each round of the algorithm satisfies $f$-DP between two Gaussian distributions. This is justified by the structure of the Sec-$f$-LDP mechanism defined in Algorithm 2, where the injected noise terms $Z_{ij,t}$ and $Z_{ji,t}$ are sampled from Gaussian distributions. As a result, the output at each iteration is a Gaussian distribution due to the linearity of gradient updates and the additive Gaussian noise. This

setup is consistent with prior analyses—e.g., [2] adopt the same assumption in their RDP-based analysis of SecDP.

Therefore, by definition, Algorithm 2 is Sec-$f$-LDP with

$$f = T(\mathcal{N}(0, \boldsymbol{\Sigma}), \mathcal{N}(\boldsymbol{\mu}, \boldsymbol{\Sigma})),$$

for some vector $\boldsymbol{\mu}$ with $\|\boldsymbol{\mu}\| = \Delta$. Here, the covariance $\boldsymbol{\Sigma}$ has the form

$$\boldsymbol{\Sigma} = \sigma_{\text{cor}}^2 \mathbf{L} + \sigma_{\text{DP}}^2 \mathbf{I}_{n-q},$$

where $\sigma_{\text{cor}}^2$ is the correlated noise, $\sigma_{\text{DP}}^2$ is the independent gaussian noise, and $\mathbf{L}$ is the Laplician matrix of the graph.

It suffices to figure out the expression of $\boldsymbol{\mu}$ in order to lower bound the above $f$ by the trade-off function between Gaussian distributions. A straightforward calculation shows that

$$T(\mathcal{N}(0, \boldsymbol{\Sigma}), \mathcal{N}(\boldsymbol{\mu}, \boldsymbol{\Sigma})) = T(\mathcal{N}(0, \mathbf{I}_{n-q}), \mathcal{N}(\widetilde{\boldsymbol{\mu}} := \boldsymbol{\Sigma}^{-1/2}\boldsymbol{\mu}, \mathbf{I}_{n-q})) = G_{\|\widetilde{\boldsymbol{\mu}}\|_2},$$

where $G_{\|\widetilde{\boldsymbol{\mu}}\|_2} = T(\mathcal{N}(0, 1), \mathcal{N}(\|\widetilde{\boldsymbol{\mu}}\|_2, 1))$ is the trade-off function for the Guassian case (i.e., $\|\widetilde{\boldsymbol{\mu}}\|$-GDP).

Note that the the eigenvalues of $\boldsymbol{\Sigma}^{-1}$ can be sorted as $\{\frac{1}{\sigma_{\text{DP}}^2 + \sigma_{\text{cor}}^2 \lambda_{n-1-i+1}\mathbf{L}}\}$ with each eigenvalue smaller than $\frac{1}{\sigma_{\text{DP}}^2}$. Moreover, the largest eigenvalue $\frac{1}{\sigma_{\text{DP}}^2}$ can be achieved since $\boldsymbol{\Sigma}^{-1}\mathbf{1} = \frac{1}{\sigma_{\text{DP}}^2}\mathbf{1}$, where $\mathbf{1}$ is a vector of ones. As a result, using the Spectral decomposition, we have

$$\|\widetilde{\boldsymbol{\mu}}\|_2^2 = \boldsymbol{\mu}^T \boldsymbol{\Sigma}^{-1} \boldsymbol{\mu} = \frac{\Delta^2}{(n-q)\sigma_{\text{DP}}^2} + \Delta^2 \sup_{\|\mathbf{x}\|=1, \mathbf{x}^T\mathbf{1}=0} \mathbf{x}^T \boldsymbol{\Sigma}^{-1} \mathbf{x}.$$

Since

$$\sup_{\|\mathbf{x}\|=1, \mathbf{x}^T\mathbf{1}=0} \mathbf{x}^T \boldsymbol{\Sigma}^{-1} \mathbf{x} \le \frac{1 - \frac{1}{n-q}}{\sigma_{\text{DP}}^2 + \lambda_2(\mathbf{L})\sigma_{\text{cor}}^2},$$

we obtain

$$\|\widetilde{\boldsymbol{\mu}}\|_2 \le \Delta\sqrt{\frac{1}{(n-q)\sigma_{\text{DP}}^2} + \frac{1 - \frac{1}{n-q}}{\sigma_{\text{DP}}^2 + \lambda_2(\mathbf{L})\sigma_{\text{cor}}^2}}.$$

$\square$

# G  Technical Details for Converting $f$-DP to $(\epsilon, \delta)$-DP

When converting $f$-DP to $(\epsilon, \delta)$-DP, particularly for addressing composition in Theorem 4.1, we employ the Privacy Loss Random Variable (PRV) and leverage numerical composition techniques as detailed in [57, 36, 29]. Recall the hockey-stick divergence $H_{e^\epsilon}(\cdot\|\cdot)$ that corresponds to the $(\epsilon, \delta)$-DP.

**Theorem G.1** (Privacy loss random variable for mixture distributions). *For mixtures distributions $P = \sum_{i=1}^m w_i P_i$ and $Q = \sum_{i=1}^m w_i Q_i$, it holds $H_{e^\epsilon}(P\|Q) \le H_{e^\epsilon}(X\|Y)$ where*

$$X|I = i \sim \log \frac{q_i(w)}{p_i(w)}, \qquad w \sim p_i,$$

$$Y|I = i \sim \log \frac{q_i(w)}{p_i(w)}, \qquad w \sim q_i,$$

*with $I$ being the indices such that $\mathbb{P}[I = i] = w_i$.*

**Remark.**  In Theorem G.1, $X$ and $Y$ are called privacy loss random variables in [29]. Thus, we can use the numerical composition method in [29] to compute the composition in Theorem 4.1.

To apply the numerical composition in [29], it is enough to clarify the CDF of $Y$. Note that $Y$ is a mixture of $\log \frac{q_i(\zeta_i)}{p_i(\zeta_i)}$, $\zeta_i \sim Q_i$ with weights $w_i$. As each $f_{ij}^t$ is a Gaussian trade-off function, both $X$ and $Y$ are mixture of Gaussian distributions.

**Corollary G.1.** *Let $P_i = \mathcal{N}(0,1)$ and $Q_i = \mathcal{N}(\mu_i, 1)$ (thus, $T(P_i, Q_i)$ is $\mu$-GDP). For the privacy loss random variable $Y$ defined in Theorem G.1, the cdf of $Y$ is $F_Y(y) = \sum_{i=1}^{n} w_i F_i(y)$ with $F_i(y) = \Phi\left(\frac{y}{\mu_i} - \frac{\mu_i}{2}\right)$. Here $\Phi$ is the cdf of a standard normal distribution.*

**Corollary G.2.** *The PRV for $f_{ij}^{\mathrm{single}}$ is $Y_{ij} \sim P_{ij}$ where $P_{ij}$ has cdf $\sum_{t=1}^{T} w_{ij}^t \Phi\left(\frac{y}{\mu_t} - \frac{\mu_t}{2}\right)$ with $\mu_t$ being defined in Lemma 4.2.*

*Proof of Theorem G.1.* Let $\xi_I$ and $\zeta_I$ be the random variable such that $\xi_I | I = i \sim P_i$ and $\zeta_I | I = i \sim Q_i$. Then, according to [61], the right-hand-side of Lemma B.1 is $T(\xi_I, \zeta_I)$ and $T(P, Q) \geq T(\xi_I, \zeta_I)$. By the Blackwell Theorem, the privacy loss of $P$ and $Q$ is bounded by that of $\xi_I$ and $\zeta_I$ and it is enough to specify the PRVs of $\xi_I$ and $\zeta_I$. We denote by $\widetilde{P}$ and $\widetilde{Q}$ the distribution of $\xi_I$ and $\zeta_I$ with pdfs $\widetilde{p}$ and $\widetilde{q}$, correspondingly. Then, according to [29], the PRV $X$ for $\xi_I$ and $\zeta_I$ is

$$X_I = \log \frac{\widetilde{q}(\xi_I)}{\widetilde{p}(\xi_I)}, \qquad \xi_I \sim \widetilde{P}.$$

Now we clarify the distribution of $X$. By the law of total probability, we have

$$\mathbb{P}[X \leq t] = \sum_{i=1}^{n} w_i \mathbb{P}[X \leq t | I = i].$$

We end the proof by noting that

$$\left. \frac{\widetilde{q}(\xi_I)}{\widetilde{p}(\xi_I)} \right| I = i$$

$$= \frac{q_i}{p_i}(\xi_i) \qquad \text{with } \xi_i = \xi_I \Big| (I = i) \sim P_i.$$

$\square$

# H  Additional Experiments

## H.1  Comparison with RDP-based Methods

**Comparison with RDP on synthetic graphs.**  To compare the performance of our $f$-DP-based account with the previous RDP-based one, we consider several synthetic (and real-life) graphs with their details in Table 2. For completeness, we visualize the synthetic graphs in Figure 4.

Interestingly, the structure of the privacy budget matrices from both methods aligns with the communicability matrix, originally introduced by [15]. Defined as $e^W$, this matrix quantifies how well-connected any two nodes are and is commonly used to detect local structures in complex networks [20]. While our PN-$f$-DP analysis does not yield a closed-form expression in terms of $e^W$, the empirical patterns suggest that the underlying graph topology plays a similar role in shaping the privacy guarantees.

| Graph name | # nodes | # edges | $\lambda_2$ | $T$ | Results |
|---|---|---|---|---|---|
| Hypercube | 32 | 80 | 0.33333 | 275 | Figure 1 |
| Cliques | 18 | 48 | 0.05634 | 1300 | Top row in Figure 5 |
| Regular | 24 | 36 | 0.10124 | 320 | Bottom row in Figure 5 |
| Expander(8) | $2^8$ | 1280 | 0.22222 | $2 \times 10^4$ | Top row in Figure 2 |
| South | 32 | 89 | 0.08209 | 110 | Figure 6 |
| Expander(11) | $2^{11}$ | 13312 | 0.16666 | $2 \times 10^4$ | Figure 7 |

Table 2: Details of the considered graphs.

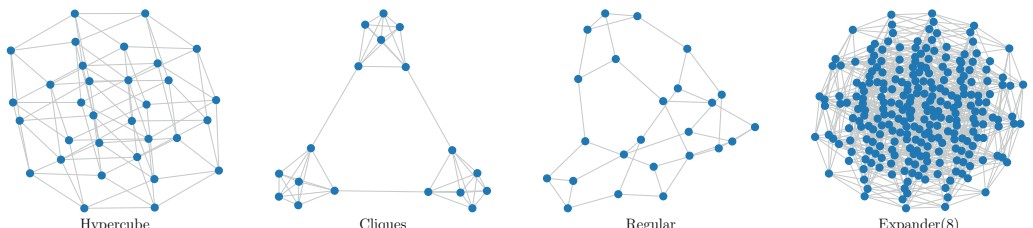

Figure 4: Visualization of synthetic graphs.

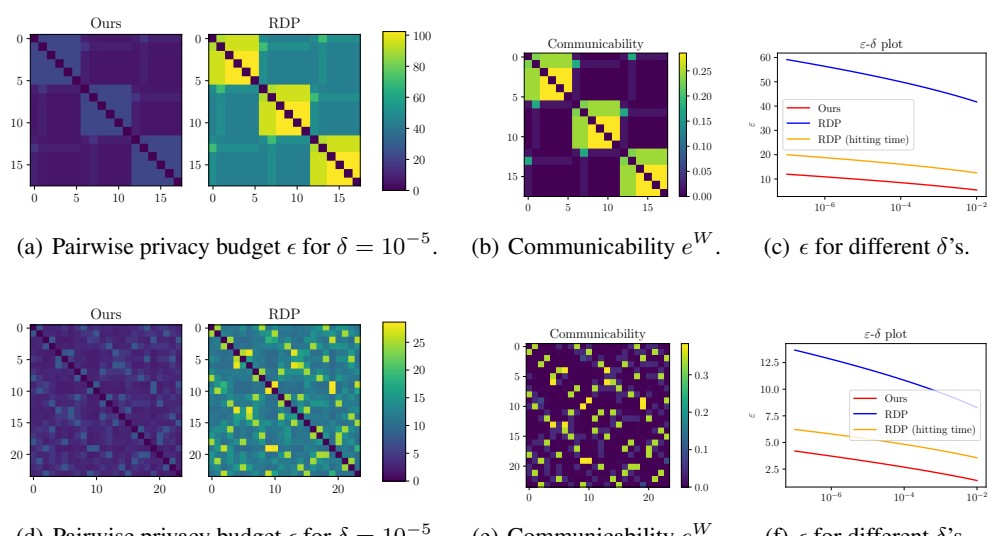

(a) Pairwise privacy budget $\epsilon$ for $\delta = 10^{-5}$. (b) Communicability $e^W$. (c) $\epsilon$ for different $\delta$'s.

(d) Pairwise privacy budget $\epsilon$ for $\delta = 10^{-5}$. (e) Communicability $e^W$. (f) $\epsilon$ for different $\delta$'s.

Figure 5: Comparison of the numerical conversion to $(\epsilon, \delta)$-DP on the clique (**top**) and regular (**bottom**) graphs.

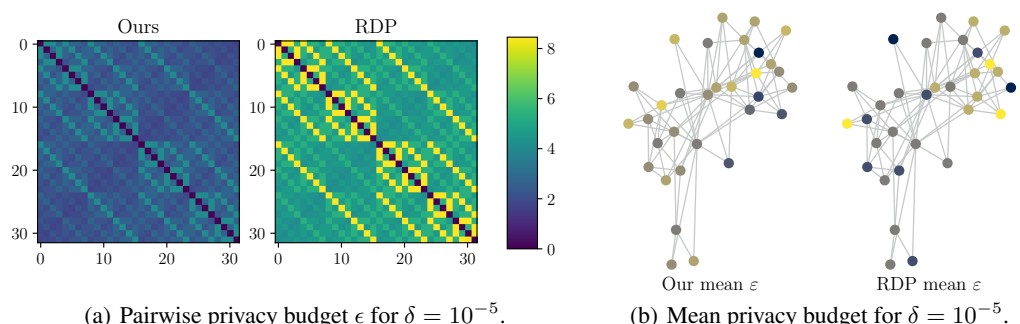

(a) Pairwise privacy budget $\epsilon$ for $\delta = 10^{-5}$. (b) Mean privacy budget for $\delta = 10^{-5}$.

Figure 6: Enhanced privacy budget by our PN-$f$-DP methods on Davis Southern women's social network.

**Privacy budget on a real graph.** Finally, we consider a real-world graph well-suited for community detection: the Davis Southern women's social network [53]. It has 32 nodes which corresponds to a bipartite graph of social event attendance by women and has been previously used by [51, 34, 15]. We present the matrix of pairwise privacy budgets $\epsilon$ in Figure 6(a) and the corresponding mean privacy budget (the average of $\epsilon_{i,j}$ across all nodes $i$ linked to node $j$) in Figure 6(b). our PN-$f$-DP privacy accounting method consistently produces a smaller privacy budget, as indicated by the lighter colors compared to the RDP results.

## H.2 Details and Results on Private Classification

**Baselines and setups.** Our work is primarily theoretical, with the main goal of introducing a general and tighter privacy accounting framework for decentralized learning. The experimental evaluation serves two purposes: (i) to validate the theoretical privacy bounds under various conditions (graph structures, noise levels, and user settings), and (ii) to ensure fair comparisons by following prior setups—particularly PN-RDP [15] and DecoR [2].

To exclude the effect of the slack parameter $\delta'_{T,n}$, we adopt the standard implementation strategy suggested in [15, Remark 5]. Specifically, once a node reaches its maximum allowed number of contributions, it stops participating in updates and only adds noise if revisited. This mechanism guarantees that the number of compositions per user never exceeds the analytical bound. In particular, when a cryptographically small $\delta'_{T,n}$ requires a very large upper bound on the number of contributions, this approach limits the actual number of communications while still preserving privacy by adding noise as needed. We apply this mechanism consistently in all PN-RDP experiments (largely because our implementations build upon their released code).

**More results on private logistic regression.** The results for a smaller network with $n = 2^8$ and $\epsilon = 10$ are presented in Figure 2. Figure 8 shows the corresponding results for $n = 2^8$ under other privacy levels $\epsilon \in \{8, 5, 3\}$, where we observe a similar pattern. The counterparts for a larger network with $n = 2^{11}$ are shown in Figure 7. All experiments are conducted on a CPU cluster with 200 GB of memory. Computing the privacy budget for all node pairs takes approximately 1–4 hours, depending on the graph structure, while the logistic regression itself completes within 2–3 minutes.

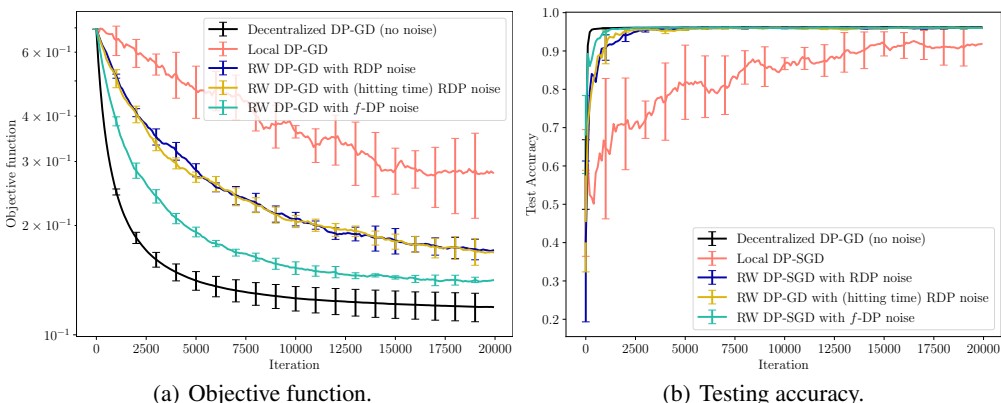

(a) Objective function.        (b) Testing accuracy.

Figure 7: Private logistic regression on the Houses dataset with $n = 2^{11}$, $L = 0.4$ and $\epsilon = 8$.

**More results on private image classification.** In addition to the previous private logistic regression experiment, we further evaluate our methods on an image classification task using the MNIST dataset [38]. The noise variance used for private training are listed in Table 3, and the evaluation follows the same privacy setting and plotting conventions.

The model we use is a simple convolutional neural network (CNN) with two convolutional layers followed by two fully connected layers. Specifically, the first convolutional layer applies 16 filters of size $8 \times 8$ with stride 2 and padding 3. The output is passed through a ReLU activation and a $2 \times 2$ max pooling layer with stride 1. The second convolutional layer applies 32 filters of size $4 \times 4$ with stride 2, again followed by a ReLU activation and another $2 \times 2$ max pooling layer with stride 1. The resulting feature maps are flattened and passed through a fully connected layer with 32 hidden units and ReLU activation, and finally mapped to 10 output logits corresponding to the digit classes.

The PyTorch implementation of the model is shown below.

```python
class MNIST_CNN(nn.Module):
    def __init__(self):
        super().__init__()
        self.conv1 = nn.Conv2d(1, 16, 8, 2, padding=3)
        self.conv2 = nn.Conv2d(16, 32, 4, 2)
        self.fc1 = nn.Linear(32 * 4 * 4, 32)
```

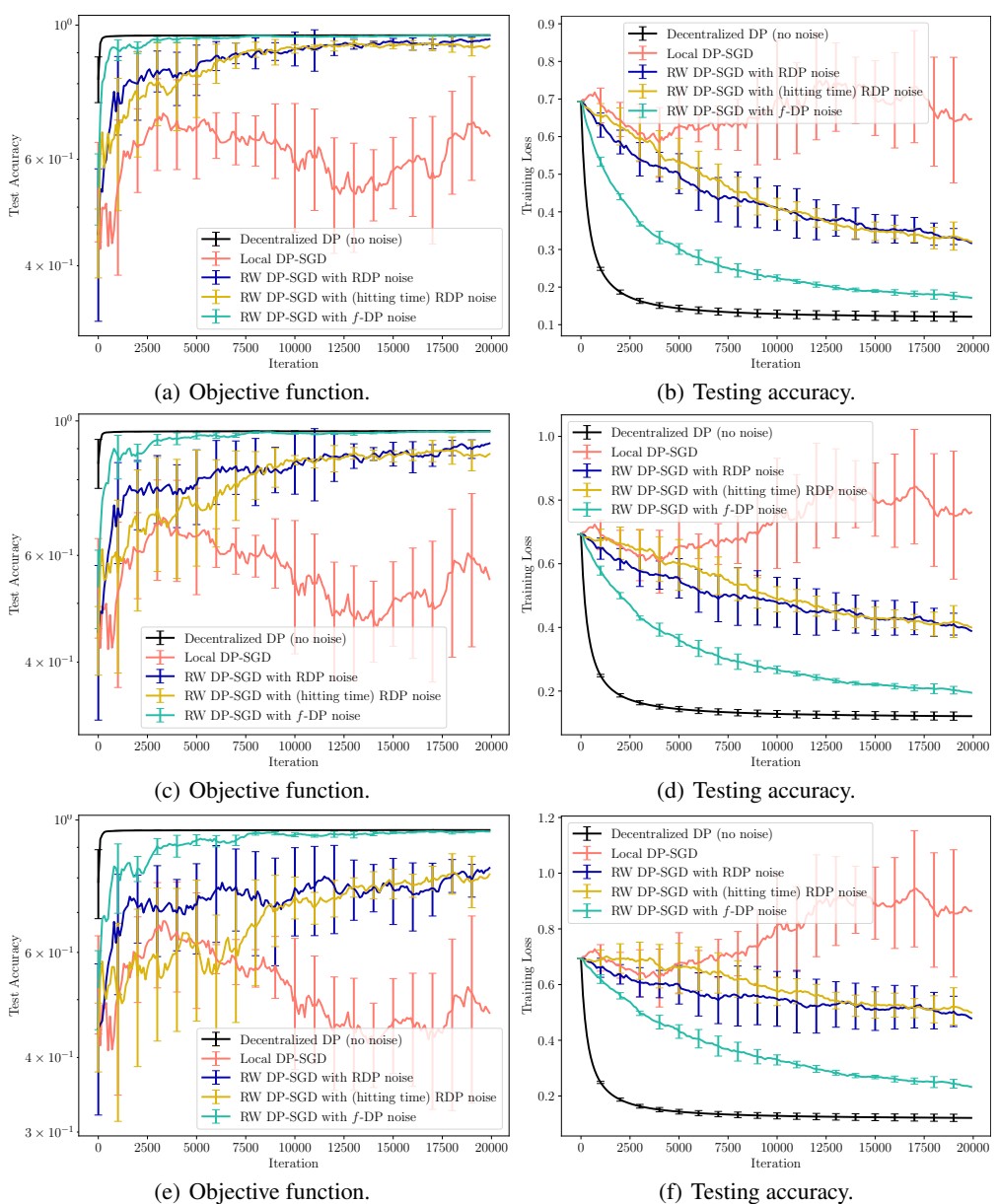

Figure 8: Private logistic regression on the Houses dataset with $n = 2^8$ and $L = 0.4$. Results are shown for $\epsilon = 8$ (**top**), $\epsilon = 5$ (**middle**), and $\epsilon = 3$ (**bottom**).

```
7            self.fc2 = nn.Linear(32, 10)
8        def forward(self, x):
9            x = F.relu(self.conv1(x))        # [B, 16, 14, 14]
10           x = F.max_pool2d(x, 2, 1)         # [B, 16, 13, 13]
11           x = F.relu(self.conv2(x))         # [B, 32, 5, 5]
12           x = F.max_pool2d(x, 2, 1)         # [B, 32, 4, 4]
13           x = x.view(-1, 32 * 4 * 4)        # [B, 512]
14           x = F.relu(self.fc1(x))           # [B, 32]
15           return self.fc2(x)                # [B, 10]
```

**More results on related noises $\sigma^2$.** For a detailed comparison, the computed noise variances are reported in Table 3. As shown, our $f$-DP–based privacy accounting method consistently requires a smaller noise variance to achieve the same differential privacy guarantee. This improvement arises

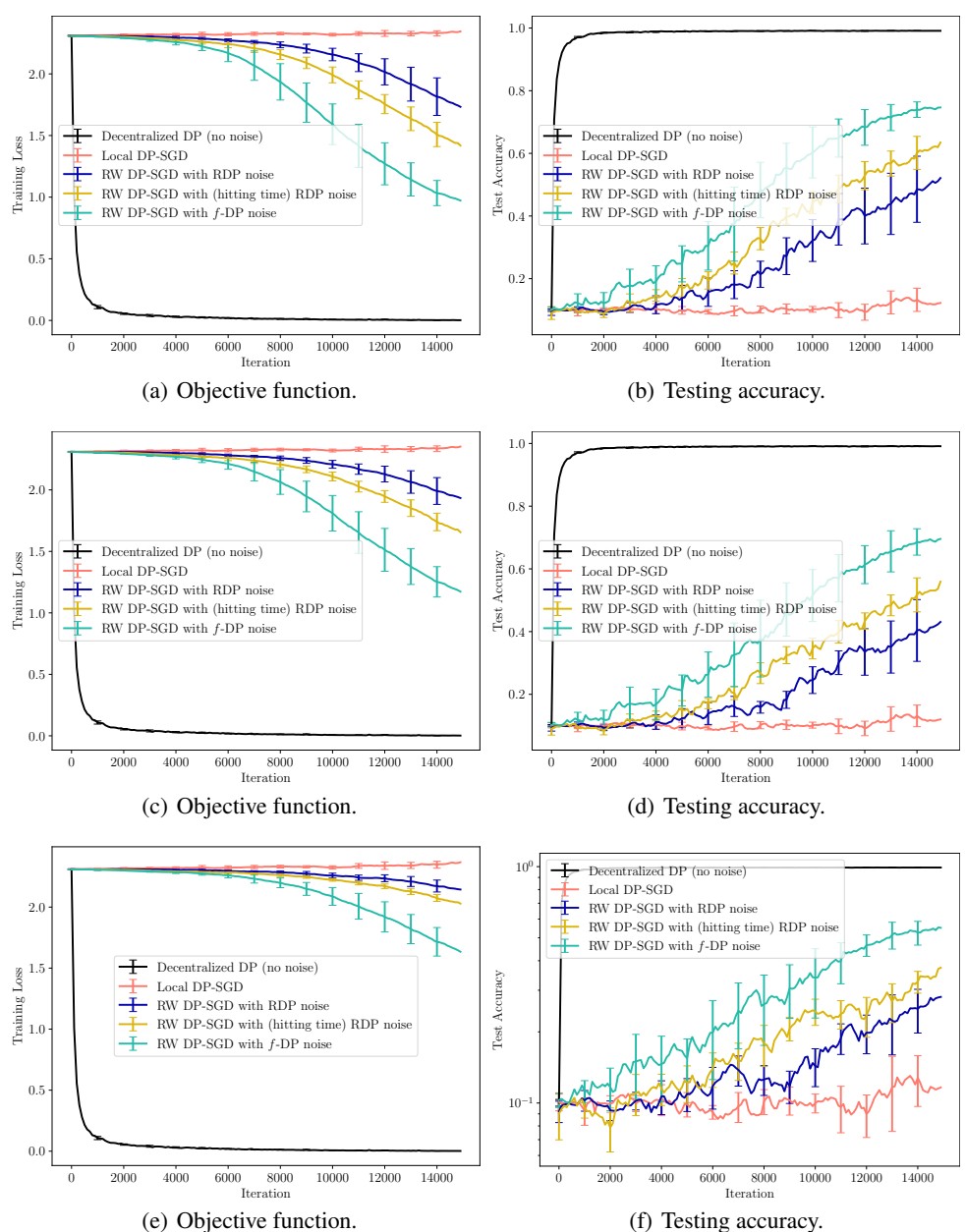

Figure 9: Private logistic regression on the MNIST dataset with $n = 2^8$ and $L = 1$. Results are shown for $\epsilon = 8$ (**top**), $\epsilon = 5$ (**middle**), and $\epsilon = 3$ (**bottom**).

from the tighter nature of $f$-DP accounting, which provides a more accurate characterization of privacy loss than RDP-based methods. Consequently, the added noise causes less degradation to the non-private algorithm, leading to better overall utility.

**Additional studies on $\delta$.** In the main text, we set $\delta = 10^{-5}$ to align with standard DP practice, where $\delta$ is typically chosen to be smaller than the inverse of the total number of data points. For example, in the MNIST dataset, which contains 70,000 samples, our choice satisfies $\delta = 10^{-5} < 1/70{,}000$. This convention is widely adopted in prior work.

One potential concern is that this choice of $\delta$ corresponds to instance-level privacy, where the replacement of a single data sample is considered. In contrast, our main analysis focuses on user-level differential privacy, a common setting in FL, where the entire local dataset of a user may be changed.

| $n$ | $\epsilon$ | $L$ | Data | Decent. DP-GD | Local DP-GD | RDP noise | RDP noise (hitting time) | $f$-DP noises |
|---|---|---|---|---|---|---|---|---|
| $2^{11}$ | 10 | 0.4 | House | 0 | 5.20637 | 0.81593 | 0.74999 | **0.32468** |
| $2^8$ | 10 | 0.4 | House | 0 | 15.8605 | 1.21531 | 0.77421 | **0.74468** |
| $2^8$ | 8 | 0.4 | House | 0 | 19.0823 | 3.62771 | 3.20279 | **0.88906** |
| $2^8$ | 5 | 0.4 | House | 0 | 28.6394 | 5.54329 | 4.89283 | **1.30494** |
| $2^8$ | 3 | 0.4 | House | 0 | 45.4803 | 8.92327 | 7.87766 | **2.01376** |
| $2^8$ | 10 | 1 | MNIST | 0 | 39.6513 | 2.98459 | 2.63431 | **1.86179** |
| $2^8$ | 8 | 1 | MNIST | 0 | 47.7058 | 3.62772 | 3.20279 | **2.22045** |
| $2^8$ | 5 | 1 | MNIST | 0 | 71.5985 | 5.54275 | 4.89298 | **3.26196** |

Table 3: Computed $\sigma$ for all involved algorithms.

In such cases, $\delta$ should instead be defined relative to the number of users rather than the total number of data samples.

To address this concern, we conducted additional experiments using $\delta = 1/$(number of users) instead of $1/$(number of total samples). This adjustment results in a larger $\delta$, effectively relaxing the privacy constraint. As expected, this leads to improved utility. The updated results, summarized in the table below (to be included in the revised manuscript), show that our method continues to perform strongly under this revised setting. Notably, RW DP-SGD with $f$-DP noise consistently outperforms all baselines, confirming its effectiveness in achieving a favorable privacy–utility trade-off.

| #Nodes | $\epsilon$ | No Noise SGD | Local SGD | RW DP-SGD (RDP) | RW DP-SGD (Hitting RDP) | RW DP-SGD ($f$-DP) |
|---|---|---|---|---|---|---|
| $2^{11}$ | 3 | 0.9614 | 0.7407 | 0.8242 | 0.7829 | **0.9534** |
| $2^{11}$ | 5 | 0.9618 | 0.8196 | 0.8912 | 0.8673 | **0.9574** |
| $2^8$ | 3 | 0.9615 | 0.5445 | 0.8328 | 0.8298 | **0.9551** |
| $2^8$ | 5 | 0.9615 | 0.5969 | 0.9004 | 0.8994 | **0.9577** |

Table 4: Comparison of test accuracies on MNIST dataset under varying $\epsilon$ and number of nodes.

## H.3 Details and Results for Correlated Noises

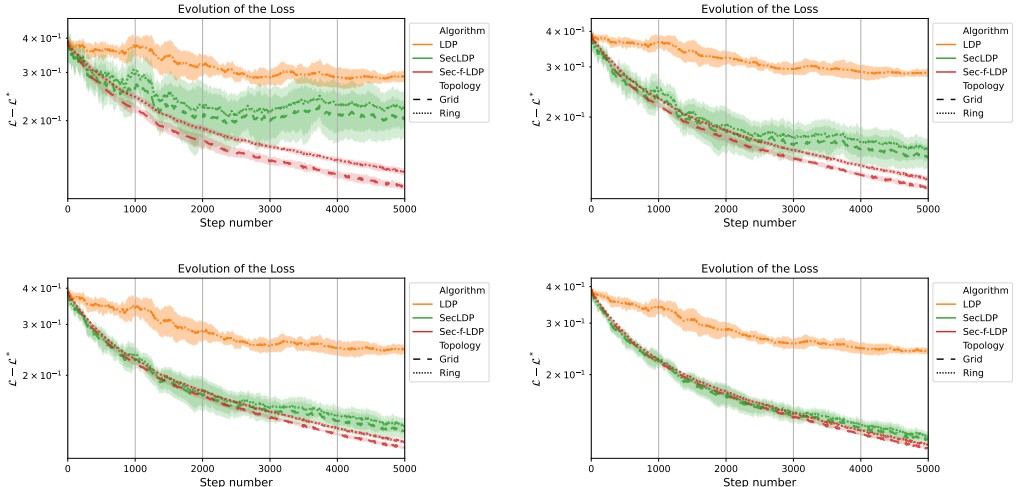

Figure 10: Optimality gap vs. iteration steps for Alg. 2 on logistic regression using different privacy accounting methods with $\epsilon =$3 (**left upper**), 5 (**right upper**), 7 (**left bottom**), and 10 (**right bottom**). Our methods perform well when the budget $\epsilon$ is small.

Our experimental setup for correlated noise follows [2]. We evaluate our method on the MNIST dataset [38], partitioned among 16 users to simulate a decentralized learning environment. Two network topologies with different levels of connectivity are considered: a ring and a 2D torus (grid).

The Metropolis-Hastings mixing matrix is utilized for model averaging across these topologies. All experiments are conducted over 5000 communication rounds for logistic regression, and 1000 communication rounds for MNIST classification. Each user updates their model parameters and exchanges information with neighbors as defined by the network topology. We repeat each experiment with four different random seeds to ensure reproducibility. We use their code in our evaluation: `https://github.com/elfirdoussilab1/DECOR`. Figure 10 shows the optimality gap versus iteration steps for Algorithm 2 on logistic regression with privacy budgets $\epsilon \in \{3, 5, 6, 7\}$. As evident from the plots, our method performs particularly well in the low-privacy regime, achieving smaller optimality gaps when the privacy budget $\epsilon$ is small.

# I  Broader Impacts

This paper proposes improved privacy accounting techniques for decentralized federated learning (FL), which can positively impact user data protection in sensitive applications such as healthcare, finance, and mobile systems. By reducing the amount of noise needed to achieve rigorous privacy guarantees, the proposed framework can help improve the utility of privacy-preserving models in real-world deployments. On the other hand, as with any privacy technology, there is a potential risk that stronger privacy accounting tools could be misused to justify under-protective practices or give a false sense of security if applied incorrectly. We encourage practitioners to use these methods with a clear understanding of their assumptions and limitations and recommend combining them with practical audits and monitoring tools in deployment.

# J  Limitations

First, the privacy analysis under $f$-DP assumes access to precise or tightly approximated trade-off functions, which can be computationally intensive to evaluate, especially for large-scale systems or long training horizons. Second, our results rely on assumptions such as fixed communication graphs and Markovian random walks, which may not hold in dynamic or adversarial network settings. Third, while we provide both user-level and record-level analyses, we do not account for adaptive adversaries or scenarios with heterogeneous trust levels beyond pairwise secret sharing. Fourth, our experiments focus on synthetic graphs and mid-scale real datasets, and results may not fully generalize to highly non-i.i.d. or large-scale production environments. Finally, our approach does not currently support other privacy mechanisms beyond additive Gaussian noise, which limits its applicability to alternative designs such as clipping-free or post-processing–based schemes.

