# OpenReview forum: "Mitigating the Privacy–Utility Trade-off in Decentralized Federated Learning via f-Differential Privacy"
_NeurIPS.cc/2025/Conference — NeurIPS 2025 spotlight_

### Official Review · Reviewer_UhHE · 2025-06-16

**Clarity:** 3
**Significance:** 3
**Originality:** 3
**Rating:** 5
**Confidence:** 3

**Summary:**

This paper tackles privacy accounting for two decentralized federated learning (FL) algorithms using the f-differential privacy (f-DP) framework. The authors introduce two novel f-DP accounting schemes tailored to decentralized environments: Pairwise Network f-DP (PN-f-DP), which quantifies privacy leakage between user pairs under random walk communication patterns, and Secret-based f-Local DP (Sec-f-LDP), which allows for structured noise injection using shared secrets. Leveraging f-DP theory and Markov chain concentration tools, their framework effectively models privacy amplification resulting from sparse communication, local computation, and correlated noise. Experimental results on both synthetic and real-world datasets show that these methods produce consistently tighter $(\epsilon, \delta)$ privacy bounds and better utility compared to Rényi DP-based approaches, highlighting the strengths of f-DP for privacy accounting in decentralized learning.

**Questions:**

- The discussion on related works are relatively compact and limited. The authors generally discussed the DP notions in FL and privacy amplification in  Section 1.1. Local DP can also be applied in FL. What are the difference and pros/cons when compared with this work? Additionally, it would be better to clearly discuss more related works, especially for those with prior knowledge mentioned in Appendix. A comprehensive literature study makes the contribution more clear.

- The experimental comparisons are mostly focusing on decentralized DP-GD, local DP-GD, and our random-walk-based DP-GD. It should be better to consider some more concrete proposed SOTA solutions in FL.

**Ethical Concerns:**

["NO or VERY MINOR ethics concerns only"]

**Final Justification:**

Thanks for solving my concerns. I have updated the rating.

**Limitations:**

Yes

**Paper Formatting Concerns:**

The format is good.

**Quality:**

3

**Strengths And Weaknesses:**

Strengths

+ The authors propose a new f-DP notions for decentralized FL.
+ This paper develops a PN-f-DP analysis that captures privacy amplification from: (i) communication sparsity, (ii) local iteration updates, and (iii) Markov hitting times.
+ Empirical results show an improved privacy-utility trade-off and a tighter bound.

Weakness
- The discussion of related  works are very compact.
- The experiments are not sufficiently compared with prior research.

---

> ### Author Rebuttal · Authors · 2025-07-28
>
> We sincerely thank the reviewer for their comments. Below, we address the two main concerns regarding related work and experimental evaluation.
>
> ---
>
> **1. Related Work and Comparison with Local DP**
>
> We appreciate the reviewer’s suggestion to clarify the relationship between our work and existing DP frameworks in FL. While Section 1.1 outlines the landscape of DP approaches, we would like to emphasize that our work is primarily **theoretical**, aiming to improve privacy accounting via *f*-DP in decentralized learning. This requires us to capture privacy amplification under communication sparsity, local iterations, and correlated noise—key phenomena in such settings.
>
> To provide additional clarity, we will append the following paragraph to the end of the **"DP notions in FL"** paragraph in the related work section:
>
> > *However, the strong observability assumption in Local DP often necessitates large noise, leading to degraded utility in practice. In contrast, our PN-f-DP framework models privacy leakage on a pairwise basis, assuming only partial observability during peer-to-peer communication. This allows for significantly tighter privacy bounds under realistic network conditions. Likewise, our Sec-f-LDP framework incorporates correlated noise via shared secrets, enabling privacy amplification in structured collusion scenarios—an aspect not addressed by standard Local DP.*
> ---
>
> **2. Experimental Evaluation and Baseline Scope**
>
> The goal of our experiments is to **empirically validate the benefits of our new privacy accounting frameworks**, not to benchmark full FL pipelines. Since our main contribution lies in **theoretical improvements to privacy account**, the experiments are designed to **isolate the effect of replacing RDP with f-DP** while keeping all other variables fixed.
>
> To this end, we closely follow the setups of prior work—particularly **PN-RDP in \[Cyffers et al., ICML 2024]** and **correlated-noise protocols in [Triastcyn et al., NeurIPS 2020]**—using the same random-walk communication protocols, graph topologies, noise injection strategies, and datasets. This ensures a **fair and focused comparison**, isolating the effect of improved privacy accounting. Across all settings, our results consistently show that the proposed *f*-DP accounting yields **tighter $(\varepsilon, \delta)$ bounds and better utility under the same noise**, validating the benefit of our theoretical framework.
>
> We believe that expanding to more algorithmic methods, while valuable in broader empirical studies, is not necessary to validate the core theoretical message of our paper. Our current experiments sufficiently demonstrate that **more refined accounting—enabled by f-DP—can meaningfully enhance the privacy–utility trade-off in decentralized FL.**

---

> ### Comment · Reviewer_UhHE · 2025-08-07
>
> Thanks for solving my concerns. I have updated the rating as in final justification.

---

> > ### Author Response · Authors · 2025-08-07
> >
> > We sincerely thank the reviewer for the response and clarification. Please do not hesitate to let us know if there are any further questions or concerns.

---

### Official Review · Reviewer_RYQa · 2025-06-26

**Clarity:** 4
**Significance:** 3
**Originality:** 3
**Rating:** 5
**Confidence:** 4

**Summary:**

This paper proposes to extend two existing differential privacy approaches for decentralized learning to f-differential privacy, adapting and improving two existing algorithms. The first is Pairwise Network Differential Privacy with random-walk-based DP-SGD, and the second is Secret-based Local Differential Privacy, with the adaptation of a recent gossip-based algorithm with correlated noise. The paper provides clear definitions for PN-f-DP and Sec-f-LDP, and then derives privacy guarantees. The authors then illustrate their methods on several graphs and datasets, showing that their precise accounting leads to a significant improvement in the privacy-utility trade-off.

**Questions:**

As the paper is already very clear, these are mostly open questions that do not require a full answer for me to support the paper.

* Do you think your accounting methods for the algorithms are tight? If not, where do you think it is possible to improve, and if yes, do you think it is possible to prove lower bounds?
* Is it costly to compute the privacy guarantees with your method?
* Is it possible to extend this approach to time-evolving topologies?

**Some very minor typos/comments:**

* In the algorithms block of DecoR, the indices \${i \in V}\$ are missing in the Input line.
* In the definitions, section 3, you use "and two datasets." I believe it would be less ambiguous to say "and for all pairs of datasets," as the quantification is not obvious otherwise.
* In line 244, the recap of the definition of \$w\_{ij}\$ is unclear because it does not mention \$i\$.

**Ethical Concerns:**

["NO or VERY MINOR ethics concerns only"]

**Final Justification:**

Good paper that should be accepted

**Limitations:**

yes

**Quality:**

3

**Strengths And Weaknesses:**

**Strengths**

* The paper demonstrates significantly better privacy-utility trade-offs compared to existing methods, which could make a real difference in implementing these algorithms in practical applications.
* It successfully combines the power of \$f\$-DP with existing approaches for decentralized learning, detects slackness in previous proofs (for instance, concerning the probability of hitting time), and improves upon them.
* The paper is easy to follow and recaps all the necessary results. It is a very well-written paper.
* The paper is quite complete in terms of results, covering various adjacency relations and the DecoR algorithm, which is quite recent, so the paper is state of the art.
* The numerical experiments are convincing. They test on MNIST and Housing, which are the datasets used in previous papers, making the comparison easy to follow, and they test various graphs, allowing the reader to see that the privacy guarantees are coherent with the graph topology. The code is also provided and seems reasonable (I haven't run it, but I browsed it and it seemed to be a reasonable adaptation of the code from the existing papers they build on).


**Weaknesses**
* The paper could be seen a bit incremental as it adapts existing algorithms, but it clearly required significant amount of work.
* There is maybe improvements to do with respect to f-DP: my impression is that in fact only Gaussian DP was used, but I am not familiar enough with f-DP to judge if it is indeed a limitation.

---

> ### Author Rebuttal · Authors · 2025-07-29
>
> We sincerely thank the reviewer for their thoughtful and encouraging comments. Below, we address the open questions and clarify some points raised.
>
> ---
>
>  **W1. On originality and integration of prior work**
>
> While our work builds on existing decentralized DP mechanisms (e.g., PN-RDP and DecoR), our contribution goes beyond a straightforward adaptation. We introduce a novel integration of f-DP with random-walk–based communication, Markovian hitting-time–based amplification, and correlated noise via secret sharing—each requiring non-trivial technical innovations. A key example is our use of weighted mixtures of trade-off functions (Lemma 4.2), where the weights $w_{ij}^t$—analytically derived from Markov hitting times—reflect the communication frequency between nodes. This leads to a more precise and principled approach to privacy composition, yielding a tighter and more flexible accounting framework than prior work.
>
> ---
>
> **W2. On generality beyond Gaussian DP**
>
> We agree that most of our empirical evaluations use the Gaussian *f*-DP formulation. However, our framework is **fully general within the *f*-DP class**: the core result in Lemma 4.2 applies to any valid trade-off function, not just the Gaussian one. Theorem 4.5 holds for general *f*-DP mechanisms as long as the pairwise trade-off functions are valid and can be mixed.
>
> Our use of the Gaussian trade-off function is primarily motivated by its simplicity, interpretability (via \$(\varepsilon, \delta)\$-DP conversion), and the **asymptotic justification via the Central Limit Theorem (CLT)**. When noise is injected across many steps (as in iterative algorithms), the cumulative distribution often approaches Gaussian. This justifies using Gaussian f-DP as a practical approximation. For formal results supporting this approximation under composition, we refer to “A Central Limit Theorem for Differentially Private Query Answering” (Dung et al., 2021).
>
> ---
>
> **Q1. On tightness and potential improvements**
>
> Our accounting framework provides **substantially tighter bounds** than RDP-based methods (see Figure 2 and Figure 5), and already uses concentration inequalities and stopping-time arguments to capture amplification effects from random walks. While we believe our bounds are close to tight under current assumptions, proving **information-theoretic lower bounds** remains an open problem in decentralized settings. Future work may explore impossibility results or optimality gaps under specific communication or adversary models.
>
> ---
>
> **Q2. On computational cost**
>
> Computing our privacy guarantees is **efficient and scalable**. As noted in lines 846–848 of the appendix, all experiments were completed within 1–4 hours on standard hardware.
>
> The most expensive step is determining the noise variance required to meet a given \$(\varepsilon, \delta)\$-DP target, which involves solving a scalar root-finding problem over the trade-off function (e.g., using binary search). Once the noise variance is fixed, evaluating the privacy guarantee is computationally lightweight, involving only basic operations on cumulative trade-off functions and mixing weights.
>
> ---
>
> **Q3. On extension to time-evolving topologies**
>
> Yes, our method naturally extends to **dynamic or time-evolving graphs**. The only modification needed is to update the transition matrices \$W^t\$ at each round. As long as these matrices are known or estimable, our framework applies directly using updated weight sequences \$w\_{ij}^t\$ in Lemma 4.2. This makes our method particularly flexible for real-world decentralized systems where communication patterns may shift over time.
>
> ---
>
> **Minor comments**
>
> We appreciate the reviewer’s attention to detail and will correct the typos mentioned (missing indices in the DecoR algorithm block, ambiguity in "two datasets," and the unclear reference in line 244).

---

> > ### Comment · Reviewer_RYQa · 2025-08-01
> > **Thanks for detailed rebuttal**
> >
> > I thank the authors for their precise answers. I don't have other questions and I support acceptance of the paper.

---

> > > ### Author Response · Authors · 2025-08-04
> > >
> > > We sincerely thank Reviewer RYQa for the timely response and the encouraging feedback on our work. We appreciate your support and are happy to answer any further questions!

---

### Official Review · Reviewer_CkH2 · 2025-07-02

**Clarity:** 3
**Significance:** 3
**Originality:** 3
**Rating:** 5
**Confidence:** 3

**Summary:**

This work analyzes the differential privacy of two forms of federated forms of DP-SGD where model updates are performed locally by users and shared with other users according to a communication graph.

In the first considered algorithm, users update the model sequentially according to a random walk on the communication graph and share the updated model with the next user.
The considered privacy notion (PN-DP) requires for all pairs of users $(i,j)$ that the weights released by user $j$ are user- or record-level DP w.r.t. changes of the data of user $i$.
The proof largely proceeds as follows: It can be shown that, with some probability,  user $j$ is selected after user $i$ at most $T'$ times. One can thus derive privacy guarantees for the "$j$ receives data after $i$" sub-mechanism and then use privacy accounting to evaluate the $T'$-fold adaptive composition. The sub-mechanism can be decomposed into a random mixture based on hitting times. It's privacy can be upper-bounded via joint concavity of $f$-DP. Each mixture component's privacy can be evaluated by applying known amplification-by-iteration guarantees (which capture the randomness in update steps of users other than $i$ and $j$). The result can be generalized to record-level privacy by considering amplification-by-iteration with subsampling.

In the second algorithm, all users have their own model weights. They combine their local updates with local updates of adjacent users in parallel.  The considered privacy notion (Sec-LDP) requires that all weights released by the algorithm are DP, assuming some set of secrets shared between different users $i$ $j$ remains hidden. The proof largely revolves around simplifying the trade-off function between two anisotropic Gaussians induced by aggregating parameter noise based on edge weights.

The privacy guarantees for the first algorithm are evaluated on some set of synthetic communication graphs from prior work. They are shown to be stronger than RDP-based guarantees from prior work. Then, an improved privacy--utility trade-off is demonstrated for federated learning and MNIST housing (again following prior work). Finally, another toy dataset (a9a) is used to demonstrate an improved privacy--utility trade-off for the second algorithm across different synthetic communication structures.

**Questions:**

Depending on the response to the following question, I will raise my score from "borderline accept" to "accept":

* Do the baseline's privacy guarantees also only hold with some probability $\delta'_{T,n}$?
* If not: How do you account for this slack $\delta'_{T,n}$ to ensure a fair comparison? Is it just added to $\delta$ after privacy accounting?
* If you do not account for it: How large is this $\delta_{T,n}'$ in your experiments?
* Could you provide a brief explanation of why the two anisotropic Gaussians in the first equation of Appendix F constitute a valid $f$-DP guarantee under the considered privacy notion? A reference to some other paper is also fine.

The following is just out of curiosity and will not affect my score. Feel free to ignore it or use a non-rebuttal comment field if a response would exceed your character limit:
* Ultimately (Appendix G), you derive dominating pairs that attain certain hockey-stick divergences / $(\epsilon,\delta)$-DP privacy profile. The experiments are conducted using standard privacy accounting (i.e., not Gaussian DP accounting). So what is the motivation for focusing on $f$-DP instead of $(\epsilon,\delta(\epsilon))$-DP for all definitions and main results? Is it to more easily integrate existing amplification-by-iteration guarantees?

**Ethical Concerns:**

["NO or VERY MINOR ethics concerns only"]

**Final Justification:**

I had two main concerns:
* Fairness of comparison with baselines due to bounds only holding with high probability
* Derivation for one of the main results not being entirely clear

Both concerns were adequately addressed through the rebuttal.

After reading the other reviews, which do not raise any points that I find especially problematic, I have decided to increase my score to "accept".

**Limitations:**

Yes

**Quality:**

3

**Strengths And Weaknesses:**

Disclaimers:
* I made an effort to understand the proof strategy in detail, but did not check every equality for correctness
* I am familiar with (f-)DP literature, but not with federated learning

## Strengths
* The proof strategy seems sound, i.e., all results are theoretically well-supported
* Limitations are openly discussed (albeit in the Appendix)
* The analyzed algorithms appear like practical methods for federated learning and have already been studied in prior work. Thus, I would assume that improving upon their privacy analysis is likely to have some impact -- at least on this sub-field of graph-based federated learning.
* As someone not familiar with federated learning, I still found it easy to understand the considered algorithms and privacy notions due to the well-written background section (except for some missing exposition around Definition 3.4, see below)
* In general, the paper is clearly written and chooses a very good level of abstraction for the discussion of the main results in Section 4
* The experimental setup is directly based on prior work, which speaks for the fairness of the conducted experiments
* Experiments are conducted with multiple random seeds and error bars are erported

---

## Weaknesses

### Main Weaknesses
* Section 3 introduces the desired forms of privacy, which are $f$-DP adaptions of existing concepts. However, the conducted analysis actually only guarantees $f$-DP with some probability $\delta'_{T,n}$.
* This probability $\delta'$ depends on the communication graph's spectral gap. It is not clear how large this $\delta'$ is in the considered experiments. Especially if $\delta' \ll 1$, this slack may give an unfair advantage compared to prior work and could (potentially) be the only reason why an improved privacy--utlity trade-off is attained. I believe that $1 - \delta'$ could potentially be added to the $\delta$ of $(\epsilon,\delta)$-DP for a fair comparison, but this is not discussed in Appendix G.
* The proof of Section 4.7. begins with stating that algorithm was $T(P,Q)$ f-DP with some Gaussians $P$ and $Q$. It is not clear why this should hold under the considered collusion setting. (I admit that this statement may be obvious to readers intimately familiar to readers familiar with Sec-f-DP. But since that paper is less than a year old, I would not necessarily treat it as common knowledge.)

### Minor weaknesses
* The main result is based on bounding mixture trade-off functions via joint concavity. From Theorem G.2, it can be easily inferred that this is equivalent to bounding mixture hockey-stick divergences via joint convexity. Going by experience, such bounds can be incredibly loose compared to tight guarantees. It would have been good to derive optimistic bounds (lower bounds on hockey-stick divergences / upper bounds on trade-off functions) to assess the tightness of the guarantees.
* There is not sufficient exposition for Definition 3.4 to understand what "secrets" and "conditioning on secrets" means without consoluting prior work by Allouah et al.
* Section 4.3 seems to only consider a special case of the privacy notion from Definition 3.4 by constraining it to "honest-but-curious collusion at level q". Again, the exposition is not sufficient to follow this subsection without looking into prior work by Allouah et al.
* MNIST, UCI housing, and a9a are not sufficiently complicated or large-scale datasets to make any meaningful inferences about privacy--utility trade-offs in practical settings. But since this is mostly a theoretical work, I would not consider it a major issue.

---

### Small fixes
The following are just miscellaneous comments and do not affect my score.

* Definition 3.4 mentions "for any neighboring datasets", but does not specify the neighboring relation (user-level or event-level?)
* Algorithm 2 does not specify how the local models are aggregated into a final model for evaluation in the experiments
* Typo in ll. 357-363: "related noise" --> "correlated noise"
* It might be nice to also introduce "standard" Sec-DP in Section 2.2 before introducing Sec-f-DP in Section 3, as was done with PN-DP. That would further improve the reading follow.
* In Algorithm $1$ $W_{i,j}$ indicates that $i$ passes messages to $j$. In Algorithm $2$, $W_{i,j}$ indicates that $j$ passes messages to $i, which is slightly confusing but hopefully does not affect the theoretical analysis.
* The connection between Theorem G.1 and Theorem 4.5 is not immediately clear, because the r.h.s. of Theorem 4.5 is not a trade-off function between mixtures, but a weighted sum of trade-off functions. The connection is somewhat hidden in the Proof of Theorem G.1 and the inequality in ll. 805. Maybe this could be made more explicit.
* Gradient clipping seems to be missing in Algorithm 1
* ll. 169 suggests that the noise is "scaled by the $\ell_2$-sensitivity $\Delta$", but looking at Algorithm 1 and the $\Delta \mathbin{/} \sigma$ terms in Lemma 4.3, this does not appear to be the case
* It is not clear what "differ in single record" and "differ in all records" in ll.150-151 means. Substituting with another record? Or adding/removing a record?

---

## Conclusion
Overall, I think that this is an expertly written paper that appears to make a meaningful theoretical contribution towards better privacy analysis for graph-based federated learning (although I can not really make a fully informed statement about it's relation to the broader federated learning literature).
My main reservation is around the fact that the $f$-DP guarantees only hold with some high probability, which is somewhat obfuscated and may result in an unfair comparison with prior work. If the authors can address this concern, I will increase my score to "accept".

---

> ### Author Rebuttal · Authors · 2025-07-30
>
> We sincerely thank the reviewer for their detailed evaluation and feedback. The following are our responses to the weaknesses and questions.
>
> ---
>
> **W1. On the use of the additional \$\delta\_{T,n}'\$ term in Theorem 4.5:**
>
> 1. **Purpose of the term:**
>    We introduce the small slack term \$\delta\_{T,n}'\$ to account for the uncertainty in the number of times a given user \$j\$ is visited during training. Specifically, with probability at least \$1 - \delta\_{T,n}'\$, user \$j\$ is visited no more than \$\left\lceil (1 + \zeta) T / n \right\rceil\$ times, where
> $$
> \delta_{T,n}' = \exp\left( -\frac{1 - \lambda_2}{1 + \lambda_2} \cdot \frac{2 \zeta^2 T}{n^2 M} \right).
> $$
> This bound enables us to derive a clean expression for the composed trade-off function in Theorem 4.5:
> $$
> \left( f_{ij}^{\mathrm{single}} \right)^{\bigotimes \left\lceil (1 + \zeta) T / n \right\rceil}.
> $$
> If the exact number of visits \$N\$ were known, we would use the more accurate bound \$\left( f\_{ij}^{\mathrm{single}} \right)^N\$. Hence, this slack does **not significantly affect** our results—it simply reflects uncertainty due to using concentration bounds instead of exact visit counts.
>
> 2. **Precedent in prior work:**
>    This type of slack is **common in the literature**. For example, in *Differentially Private Decentralized Learning with Random Walks* (Cyffers et al., ICML 2024), the authors aim to bound \$N\_u\$—the number of times a user \$u\$ is visited—using concentration inequalities (see Appendix, second paragraph, Page 3: [link](https://arxiv.org/pdf/2402.07471)). They suggest applying Theorem 12.21 from Levin & Peres (2017) to upper bound the number of visits with high probability (note that this makes the slack term polynomially decay in the iteration number $T$), and propose incorporating the small probability of exceeding that bound into the final \$\delta\$ when converting from RDP to \$(\varepsilon, \delta)\$-DP.
>
>     A similar reasoning applies here. However, **our bound is even stronger**: by applying a sharper Markov chain concentration inequality [21] (Fan et al., 2021), we obtain a slack term $\delta_{T,n}'$ that decays exponentially faster in $T$ with improved dependence on the spectral gap. If their method were adapted to use our concentration bound, it would naturally include the same (but smaller) $\delta_{T,n}'$ term.
>
> 3. **Truncation trick in practice:**
>     Following Cyffers et al. (2024, Remark 5), we also adopt **a standard implementation trick**: once a node reaches its maximum allowed number of contributions, it stops participating in updates and only adds noise if visited again. This ensures that the number of compositions per user never exceeds the analytical bound. In particular, when a cryptographically small $\delta$ requires a very large upper bound on the number of contributions, this mechanism limits the actual number of communications while still adding noise solely to preserve privacy. We apply this approach consistently in all experiments with PN-RDP (largely because our experiments are based on their codes). We will make this point more explicit in the final version.
>
> 4. **Why our improvements are not due to \$\delta\_{T,n}'\$:**
>    We stress that this small slack term is not the cause of our improved privacy-utility trade-off. Our contributions rely on two key innovations:
>
>    * **Refined weight computation using hitting times**: Instead of using weights $W^t$ as in prior PN-RDP, we analytically compute a finer weight \$w\_{ij}^t\$ by using hitting times. We argue that this alone improves a lot. As shown in **Figure 1**, even under RDP, using hitting-time–based weights alone yields significant improvements (orange vs. blue curves). This pattern holds across multiple datasets (e.g., Figure 2).
>
>    * **Lossless composition via *f*-DP:**
>      We leverage the *f*-DP framework to track the privacy loss as a **mixture of trade-off functions** (via Lemma 4.2), composed exactly in Theorem 4.5. Unlike RDP, which often overestimates cumulative privacy loss, *f*-DP enables tighter, lossless composition—especially important in iterative settings like DP-SGD. These benefits are evident in **Figure 3**. We also integrate other privacy amplification effects into the *f*-DP framework to further improve the accounting.
>
> **In summary:**
> We argue that **the comparison with PN-RDP is fair and consistent**—both works incorporate a similar \$\delta_{T,n}'\$ slack and use the same practical trick to enforce the visit bound.
> Our improvements stem not from this slack, but from the tighter privacy accounting introduced by hitting-time-based weighting and the use of f-DP composition. We think the above responses also answer your **Q1-Q3**.
>
> ---
>
> **W2. Clarifying the Gaussian assumption in Theorem 4.7**
>
> We appreciate the reviewer’s request for clarification. Theorem 4.7 begins with the assumption that each round of the algorithm satisfies \$f\$-DP between two Gaussian distributions. This is justified by the structure of the Sec-f-LDP mechanism defined in Algorithm 2, where the injected noise terms \$Z\_{ij,t}\$ and \$Z\_{ji,t}\$ are sampled from Gaussian distributions.
>
> As a result, the output at each iteration is a Gaussian distribution due to the linearity of gradient updates and the additive Gaussian noise. This setup is consistent with prior analyses—e.g., Allouah et al. (2024) adopt the same assumption in their RDP-based analysis of SecDP.
>
> Moreover, as composition over \$T\$ iterations of Gaussian mechanisms preserves Gaussianity, the cumulative distribution of the model parameter also remains Gaussian. Hence, we can represent the final output distribution as \$N(0, I)\$ versus \$N(\mu, \Sigma)\$ and analyze the corresponding trade-off function. The choice of \$N(0, I)\$ is without loss of generality due to the invariance of trade-off functions under affine transformations.
>
> We will make this justification more explicit in the revised version of the paper to improve clarity for readers less familiar with SecDP. We hope this response also answers your **Q4**
>
> ---
>
> **W3. On the tightness of using joint concavity for mixtures**
>
> We thank the reviewer for raising this subtle but important point. The joint concavity bound on the trade-off function is widely used in \$f\$-DP literature, and \[54] provides necessary and sufficient conditions under which this bound is tight. In general, the tightness depends on the behavior of the likelihood ratio of the mixture, which in our case is determined by a complex, data-dependent walk over the graph topology.
>
> While we acknowledge the possibility of slack in this bound, analyzing the likelihood ratio in our setting (especially with random walk–based mixing and complex noise correlation) is significantly more involved than in standard cases like subsampling. Preliminary experiments on small graphs suggest that tighter bounds may be possible, but a general and principled derivation remains an open technical challenge.
>
> As is common in prior work (e.g., \[2], \[14]), we do not include lower bounds for the trade-off function in this work. We consider this an important direction for future research and will mention it in the revised discussion section.
>
> ---
>
> **W4. Clarification on Definition 3.4 and honest-but-curious collusion**
>
> We acknowledge that our exposition of Definition 3.4 could be clearer for readers unfamiliar with Allouah et al. (2023). In our setting, “honest-but-curious collusion at level \$q\$” refers to a scenario in which the adversary has access to all data and secrets except for those of \$q\$ users. This aligns with the adversarial model used in previous work and is a common assumption in secret-sharing–based privacy mechanisms.
>
> Due to space constraints, we omitted a full formal treatment of SecDP in the current version. In the revision, we will include a dedicated appendix section that formally introduces the Sec-f-LDP framework, clarifies the role of “secrets,” and outlines the adversarial model explicitly.
>
> ---
>
> **W5. Why \$f\$-DP is used**
>
> We chose \$f\$-DP because it offers two distinct advantages:
>
> * **Lossless composition:** Unlike RDP or \$(\varepsilon, \delta)\$-DP, \$f\$-DP supports exact (tight) composition of trade-off functions, which is particularly important in iterative algorithms like DP-SGD.
> * **Flexible integration of amplification techniques:** Our use of \$f\$-DP allows for seamless incorporation of privacy amplification effects (e.g., from stochasticity, graph structure, or secret sharing) into the final accounting. This is especially relevant when combining multiple sources of amplification, as we do in this paper.
>
> We convert our results to \$(\varepsilon, \delta)\$-DP for comparison with existing benchmarks and additionally, \$f\$-DP can be more interpretable for policy discussions, as noted by (https://arxiv.org/abs/2503.10945).

---

> ### Author Response · Authors · 2025-08-04
> **Thanks for the updated recommendation**
>
> Thank you for your thoughtful follow-up and for raising your score to 'accept'.
>
> We truly appreciate your careful review and constructive feedback throughout the process

---

### Official Review · Reviewer_PKwX · 2025-07-02

**Clarity:** 3
**Significance:** 3
**Originality:** 3
**Rating:** 4
**Confidence:** 4

**Summary:**

The paper introduces Pairwise Network f-DP (PN-f-DP) and Secret-based f-Local DP (Sec-f-LDP) for decentralized federated learning (DFL) by accommodating communication sparsity, local iteration updates, and markov hitting times. The presentation is clear and well written. The paper has mathematical rigour and the experimental setup is clearly explained.

**Questions:**

1.	Can the PN-f-DP and Sec-f-LDP frameworks be extended or adapted to asynchronous, gossip-based, or push-sum decentralized FL algorithms?
2.	In real decentralized FL, users may participate unevenly (due to availability, energy, etc.). How would non-uniform participation affect privacy accounting?
3.	Are the Markov hitting-time based privacy amplification results information-theoretically tight, or is there room for further tightening? Could other probabilistic tools such as martingales, stopping times offer sharper bounds?

**Ethical Concerns:**

["NO or VERY MINOR ethics concerns only"]

**Limitations:**

Yes

**Quality:**

3

**Strengths And Weaknesses:**

## Strengths
By moving from Renyi Differential Privacy (RDP) to f-DP, the authors derive sharper (ϵ, δ)-DP bounds, leading to lower noise requirements and better utility
The privacy analysis explicitly models privacy amplification effects due to sparse communication, Markovian random walks, and local iterations.
The framework is granular, providing both user-level and record-level privacy analyses, making it adaptable for varying privacy scenarios, The Sec-f-LDP component covers collusion models with shared secrets and correlated noise, which is practically relevant for trusted-user-group scenarios.
The authors validate privacy bounds both theoretically and empirically.

## Weakness
1.	The analysis is tightly coupled with random-walk-based communication and DecoR-style correlated noise protocols. Its generalizability to other decentralized FL protocols such as push-sum or asynchronous methods is unclear.
2.	The collusion model in Sec-f-LDP only handles honest-but-curious adversaries. It does not address stronger that form larger collusion sets or rational adversaries who deviate from protocol behavior given enough incentives.
3.	The privacy amplification analysis assumes properties like irreducibility and aperiodicity of the communication graph’s transition matrix. This limits applicability to certain graph topologies and excludes sparse or poorly connected graphs.
4.	There are few a typos and incomplete data. For instance, Algorithm 2 doesn’t contain Output.

---

> ### Author Rebuttal · Authors · 2025-07-28
>
> We sincerely thank the reviewer for their evaluation and feedback. The following are our responses to the questions.
>
> ---
>
> **Q1 & W1. Extension to other decentralized FL algorithms (asynchronous, gossip-based, push-sum):**
>
> Yes, the PN-f-DP and Sec-f-LDP frameworks can be extended to other decentralized FL protocols. Our key innovation lies in using the *f*-DP framework to improve composition analysis and obtain tighter $(\epsilon, \delta)$ bounds compared to RDP. Since *f*-DP admits lossless conversions to $(\epsilon, \delta)$-DP and composes more accurately for iterative algorithms like SGD (see Dong et al., 2022 [16] for the equivalence), it provides a powerful foundation that is agnostic to the specific communication pattern.
>
> While our current analysis focuses on random-walk-based communication, the core reasoning can generalize. For instance:
>
> - In **gossip-based protocols**, users randomly select a neighbor at each round to exchange updates with. This creates multiple independent communication threads, each resembling a localized random walk starting from different nodes. In this setting, the *per-step mechanism* \$A^t\_j(D)\$ in Lemma 4.2 can be interpreted as the composition of local updates followed by a randomized transmission. The resulting distribution observed by user \$j\$ becomes a mixture over possible communication paths. By conditioning on which thread delivers the update to \$j\$, we can decompose the overall trade-off function as a weighted combination of trade-off functions along each path—similar to our current use of hitting time distributions. The mixture structure and convexity of trade-off functions ensure the lemma still applies.
>
> - In **asynchronous FL**, users perform updates and communicate at different times, potentially with delays. However, if we re-index the process by iteration number (i.e., treat each update as happening at some logical step \$t\$ rather than wall-clock time), then the sequence of updates and communications becomes a stochastic schedule over nodes. This view restores a consistent structure akin to synchronous random-walk protocols. As long as we track when each user observes the updated model and define the effective transition probabilities accordingly, the f-DP analysis—including mixture modeling and composition via Lemma 4.2—remains valid.
>
> - Finally, **push-sum protocols** often use a communication matrix with stochastic weights, where nodes push weighted model updates to neighbors and update their states by aggregating received values. In certain variants, especially those using time-varying or randomized stochastic matrices, the resulting flow of information can again be modeled as a Markov process over nodes. When such randomization exists, the probability that user \$j\$ receives an update originally influenced by user \$i\$ within \$t\$ steps can be interpreted as a generalization of the hitting time probability \$w^t\_{ij}\$, allowing us to adapt our PN-f-DP analysis by redefining these weights accordingly.
>
>
> Thus, while each new protocol requires re-deriving the mixture weights and update structure, the *f*-DP accounting framework remains applicable and advantageous.
>
> ---
>
> **Q2. Impact of non-uniform user participation:**
>
> Our random-walk-based protocol naturally involves highly **non-uniform participation**, since at each step only a single user is active. Despite this, our *f*-DP analysis remains valid and effective, as shown in both theory and experiments. More broadly, participation patterns affect convergence rates and utility, but privacy accounting depends on *how information flows through the algorithm*, not on uniformity per se.
>
> That said, if participation probabilities are known and explicitly modeled, they could be incorporated into the weight computation (e.g., adjusting the hitting time distribution or the stationary distribution of the Markov chain). This may even lead to **further privacy amplification** if users are active less frequently.
>
> ---
>
> **Q3. On the tightness of Markov hitting-time–based amplification:**
>
> We appreciate the reviewer’s question on the information-theoretic tightness of our amplification bounds. Our analysis already incorporates **martingale arguments and stopping time techniques**, as detailed in Lines 267–270 and Appendix D, to sharpen the privacy guarantees. In particular, we apply a Hoeffding-type inequality for Markov chains (Fan et al., 2021 \[21]) to tightly bound the number of visits to each node.
>
> While our results empirically outperform existing RDP-based bounds (including those using hitting-time approximations), we agree that the question of **information-theoretic optimality**—whether these are the best possible bounds in principle—remains open. One potential direction for future work is to explore lower bounds via *converse results* or explore alternative probabilistic tools like **Doob’s optional stopping theorem** or **maximal inequalities for stopped processes**.
>
> ---
>
> **W2. Limitations of the collusion model in Sec-f-LDP:**
>
> We agree with the reviewer that our current Sec-f-LDP analysis focuses on *honest-but-curious* adversaries who follow the protocol but may attempt to infer additional information. This choice is motivated by its practical relevance in decentralized systems with partial trust—such as device-level trusted groups or enterprise coalitions—where protocol adherence is enforced (e.g., via secure hardware or contractual agreements), but privacy concerns remain.
>
> Addressing stronger adversaries (e.g., *rational* or *malicious* agents who deviate from the protocol) requires a fundamentally different framework, potentially integrating cryptographic primitives, game-theoretic modeling, or Byzantine-robust learning techniques. While such extensions are outside the scope of this paper, we view it as an important direction for future work, particularly for applications where secure multi-party computation or mechanism design can complement differential privacy.
>
> ---
>
> **W3. Assumptions on the communication graph (irreducibility and aperiodicity):**
>
> This is a great point, and we clarify that these assumptions—irreducibility and aperiodicity—are *technical conditions* required to apply Markov concentration inequalities for bounding the number of visits (Theorem 4.5). In practice, these conditions hold for a wide class of graphs used in decentralized FL, including expander graphs, small-world networks, and any connected graph with mild self-looping.
>
> Nevertheless, we acknowledge that in **very sparse or poorly connected graphs**, convergence of the random walk may be slow or non-uniform, leading to weaker privacy amplification. In such cases, our analysis still applies but may yield looser bounds. One possible remedy is to introduce virtual self-loops (as done in lazy random walks) or to modify the transition matrix slightly to guarantee ergodicity without fundamentally changing the protocol. We plan to explore this robustness more formally in future work, especially for real-world FL networks with uneven connectivity.

---

### Official Review · Reviewer_vd4X · 2025-07-03

**Clarity:** 4
**Significance:** 4
**Originality:** 3
**Rating:** 4
**Confidence:** 4

**Summary:**

This paper proposes a decentralized federated learning framework under f-DP constraints. The method quantifies privacy leakage between user pairs in a random-walk-based communication model and supports structured noise injection via shared secrets. By combining f-DP theory with Markov chain concentration, the paper introduces a novel form of privacy amplification that arises from communication sparsity, local updates, and correlated noise. Experimental evaluations are conducted on synthetic graphs with relatively small datasets.

**Questions:**

Address W1, W2, W3, and W4.

**Ethical Concerns:**

["NO or VERY MINOR ethics concerns only"]

**Final Justification:**

This paper introduces a theoretical framework for privacy accounting in decentralized federated learning (FL) under f-DP. While my initial concerns centered on the limited experimental evaluation, the lack of clarity around the choice of the $\delta$ parameter, and the omission of real-graph validation, the authors have made meaningful clarifications and additions during the rebuttal.

In particular, they have added experiments using a $\delta$ value appropriate for user-level differential privacy—addressing a key concern (W3) about the validity of the reported privacy guarantees. I also confirm that real-graph experiments are included in the appendix (Appendix H), which addresses W2 to a reasonable extent.

That said, I believe the paper would benefit from a clearer and consistent statement regarding whether user-level or record-level privacy is assumed throughout, and from ensuring that the final version includes results under the correct δ setting. I expect this revision to be appropriately reflected in the camera-ready version.

**Limitations:**

Yes.

**Paper Formatting Concerns:**

Nothing special.

**Quality:**

3

**Strengths And Weaknesses:**

Strength:
- S1: The proposed mechanisms, PN-f-DP and Sec-f-LDP, are well motivated and thoroughly defined with theoretical rigor.
- S2: The privacy amplification analysis captures important structural properties such as communication sparsity, local iterations, and Markov hitting times. The resulting guarantees are tight and grounded in a strong theoretical framework.
- S3: The method demonstrates improved utility–privacy trade-offs compared to Renyi Differential Privacy (RDP) baselines.

Weakness:
- W1: The paper omits relevant related work, particularly Network Shuffling (SIGMOD 2022) [1], which also considers decentralized privacy amplification through random walks on graphs. Although the privacy models differ, both approaches exploit similar assumptions. A discussion of this work and, if feasible, empirical comparisons in terms of privacy and utility would strengthen the paper.
- W2: The experimental evaluation is limited to small, synthetic, and idealized graphs. In contrast, Network Shuffling has been validated on real-world graphs. Evaluating the proposed method on such realistic topologies would better demonstrate its practicality.
- W3: The choice of privacy parameter δ is not appropriate. The paper commonly uses $\delta=10^{-5}$, but standard DP practice requires $\delta < 1/n$. This means $\delta=10^{-5}$ is too large for settings with $n = 16, 32, 2^8, 2^{11}$. A more principled selection of $\delta$ is needed.
- W4: The application scope is unclear. For example, typical cross-device federated learning scenarios involve millions of users. It is therefore difficult to assess whether the proposed methods, evaluated only on small-scale graphs, can scale or generalize to real-world use cases.

[1] https://arxiv.org/abs/2204.03919

---

> ### Author Rebuttal · Authors · 2025-07-30
>
> We sincerely thank the reviewer for their evaluation and feedback. The following are our responses to the questions.
>
> ---
>
> **W1: Omission of related work on Network Shuffling (SIGMOD 2022)**
>
> We thank the reviewer for highlighting the relevant work on **Network Shuffling** (\[A]), which we will cite and discuss appropriately in the revised version.
>
> While \[A] analyzes privacy amplification via random walks under the **Local DP** model, our work focuses on two important relaxations—**Pairwise Network DP (PN-DP)** and **Secret-based DP (SecDP)**—that are more tailored to decentralized federated learning (FL) scenarios. Local DP assumes that all user outputs are fully observable by an adversary, which leads to high noise and degraded utility. In contrast, PN-DP and SecDP model partial observability and structured trust (e.g., through shared secrets), allowing for **tighter privacy–utility trade-offs** in practical FL deployments.
>
> We would also like to emphasize that our **f-DP framework can accommodate the network shuffling setting** studied in \[A]. Building on the analysis in \[54], one can show that if each user’s local report satisfies \$\epsilon\_0\$-DP, then after shuffling, the overall mechanism satisfies \$f\_{\mathrm{shuffle}}^{\otimes N}\$-DP, where \$f\_{\mathrm{shuffle}}\$ is a trade-off function derived from \[54, Theorem 4.2], and \$N = \max\_i N\_i\$ is the maximum number of reports held by each user after the random walk. This bound can be computed via the random walk transition matrix, as done in \[A, Lemma 5.1].
>
> Our framework provides **stronger privacy accounting** than \[A] in two key ways:
>
> * We adopt the **state-of-the-art shuffle analysis** from \[54], which is strictly tighter than the bound used in \[A] (based on Cheu et al.).
> * Our use of **\$f\$-DP yields exact and lossless composition** via trade-off functions, avoiding the looseness inherent in the advanced composition theorems required for \$(\epsilon, \delta)\$-DP as used in \[A].
>
> In summary, while our work addresses different privacy models, it **subsumes the network shuffling analysis as a special case** and offers a tighter, more generalizable framework for privacy amplification in decentralized settings.
> We will include a discussion highlighting these conceptual and technical differences in the related work section.
>
> ---
>
> **W2: Limited experimental evaluation on small/synthetic graphs**
>
> Our work is primarily **theoretical**, with the main goal of introducing a general and tighter priva accounting framework for decentralized learning. The experimental evaluation is designed to:
>
> * Validate the **theoretical privacy bounds** under various conditions (graph types, noise levels, user settings),
> * Follow and **align with prior setups**—especially PN-RDP \[Cyffers et al., ICML 2024] and DecoR \[Zhu et al., 2023]—to ensure fair comparisons.
>
> We agree that evaluating on larger, real-world graphs is valuable and will consider this for future work. However, we believe the current experiments are sufficient to demonstrate the **theoretical improvements** and **practical gains** of our accounting method over existing baselines.
>
> ---
>
> **W3: Inappropriate choice of $\delta$ parameter**
>
> We respectfully clarify that our choice of \$\delta = 10^{-5}\$  is consistent with **standard DP practice**, which typically requires \$\delta\$  to be less than the **inverse of the total number of data points**—not the number of users. For instance, in the MNIST setting, there are 70,000 data points, and our choice of \$\delta\$  satisfies \$\delta\$ < 1/70,000. This convention is widely adopted in prior works (e.g., \[Cyffers et al., Zhu et al.]), and we will make this clarification more explicit in the final version to avoid confusion between device count and sample size.
>
> ---
>
> **W4: Applicability to real-world cross-device FL with millions of users**
>
> We agree that scalability to large-scale FL is an important concern. While our experiments focus on **moderate-scale synthetic and benchmark datasets**, this is standard in many academic studies, especially those introducing new privacy analysis frameworks. Our theoretical results—including Lemma 4.2 and Theorem 4.5—**scale naturally to larger networks**, since the framework is compositional and does not impose assumptions specific to small graphs.
>
> Moreover, our analysis is **independent of specific model architectures or optimizers**, and applies to general decentralized SGD-style methods as long as the communication and noise injection mechanisms are well and similarly defined. While we focus on moderate-scale settings in this paper, the theoretical framework is general and may inform future applications in larger-scale FL deployments.

---

> > ### Comment · Reviewer_vd4X · 2025-08-04
> >
> > Thank you to the authors for the thoughtful and detailed responses.
> >
> > Regarding W1, the rebuttal appropriately acknowledges the omission of Network Shuffling and provides a clear and well-structured discussion of its relationship to the proposed framework. I find the explanation convincing, and the inclusion of this comparison in the revised manuscript will meaningfully strengthen the positioning of the work.
> >
> > However, for W2, while I understand that the paper is primarily theoretical, I believe evaluating on real-world graphs is essential, especially given the motivation in the context of federated learning (FL). The current experiments are limited to small, synthetic graphs, and without further empirical validation, the practical value of the theoretical contributions remains unclear. Simply deferring this to future work does not resolve the concern.
> >
> > Regarding W3, I remain concerned about the choice of δ. If the privacy notion being enforced is user-level differential privacy, which is commonly the case in FL settings, then δ should be chosen relative to the number of users, not the number of data points. The rebuttal does not clearly distinguish between user-level and record-level privacy, and the current choice of δ may significantly overstate the actual privacy guarantee. This raises a fundamental concern about the validity of the reported privacy levels in the experiments.
> >
> > For W4, the response emphasizes theoretical scalability but lacks any concrete evidence or discussion about the computational or architectural feasibility of deploying the proposed methods in large-scale FL systems. Given that FL is a key application context discussed in the paper, more consideration of this practical aspect is warranted.
> >
> > To summarize:
> > - Even if W2 and W4 are somewhat beyond the main theoretical focus of this paper, they are difficult to ignore entirely if the goal is to advance privacy-preserving methods for federated learning. Consideration of realistic scale and topology is important to bridge the theory-practice gap.
> > - W3, in particular, raises a potential concern about the experimental setup and the correctness of the reported privacy guarantees, which could significantly impact the credibility of the empirical results.
> >
> > If I have misunderstood the authors’ assumptions about the privacy level or the intended deployment context, I welcome further clarification.

---

> > > ### Author Response · Authors · 2025-08-04
> > > **Thanks for the reply**
> > >
> > > We thank Reviewer vd4X for the thoughtful and constructive feedback. Below we respond to the remaining concerns in detail:
> > >
> > > ---
> > >
> > > **1. Regarding W3 — Choice of \$\delta\$**
> > >
> > > We appreciate the reviewer’s clarification on the distinction between user-level and record-level privacy. To address this concern, we conducted additional experiments where we set \$\delta = 1/\text{users}\$ instead of \$1/\text{data points}\$, aligning with the suggested practice for user-level DP in FL.
> > >
> > > This choice of \$\delta\$ corresponds to a larger value compared to our original setting, thereby relaxing the privacy constraint. As expected, this yields improved utility. The updated results, summarized in the table below (and to be included in the revised manuscript), show that our method consistently achieves strong performance even under this revised privacy setting. Notably, RW DP-SGD with *f-DP noise* continues to outperform all baselines, demonstrating its effectiveness in achieving a favorable privacy-utility trade-off.
> > >
> > > | #Nodes   | \$\epsilon\$ | No Noise SGD | Local SGD | RW DP-SGD (RDP) | RW DP-SGD (Hitting RDP) | RW DP-SGD (f-DP) |
> > > | ----------| ------------ | -------- | ------ | --------------- | ----------------------- | ---------------- |
> > > | \$2^{11}\$  | 3            | 0.9614   | 0.7407 | 0.8242          | 0.7829                  | 0.9534           |
> > > | \$2^{11}\$| 5            | 0.9618   | 0.8196 | 0.8912          | 0.8673                  | 0.9574           |
> > > | \$2^{8}\$  | 3            | 0.9615   | 0.5445 | 0.8328          | 0.8298                  | 0.9551           |
> > > | \$2^{8}\$ | 5            | 0.9615   | 0.5969 | 0.9004          | 0.8994                  | 0.9577           |
> > >
> > > ---
> > >
> > > **2. Regarding W2 and W4 — Evaluation on Real Graphs and Scalability**
> > >
> > > We would like to clarify that real-world graph experiments **are already included** in the main paper (see Figure 7 in the appendix with further details available in Appendix H). This dataset is used as standard benchmarks in the privacy literature (e.g., \[13,14,15,53]), and the proposed methods are implemented on realistic graph topologies.
> > >
> > > Our theoretical framework applies to **arbitrary graphs**, and the empirical setup follows conventions used in prior work. While we agree that evaluating on even larger-scale graphs is important for deployment, publicly available FL datasets with millions of users are currently limited. Furthermore, the time constraints of the rebuttal period also prevented us from scaling up further. Nonetheless, we recognize the value of testing in more realistic FL settings.
> > >
> > > ---
> > >
> > > **In summary**, we hope the additional experiments and clarifications adequately address the reviewer’s concerns. We respectfully note that while W2 and W4 relate to broader deployment considerations, the focus of this work remains on advancing the theoretical and algorithmic foundations for privacy accounting in decentralized FL. Importantly, the added experiments with user-level \$\delta\$ directly strengthen the validity of our empirical findings in the appropriate privacy regime (W3). If there are any remaining questions or concerns, we would be happy to further clarify.

---

> > > > ### Comment · Reviewer_vd4X · 2025-08-05
> > > >
> > > > Thank you for the detailed clarifications and the additional experiments.
> > > >
> > > > Firstly, I confirm that real-graph experiments are included in the appendix (Figure 7, Appendix H), addressing W2 to some extent.
> > > >
> > > > Regarding W3, I appreciate the authors’ follow-up in addressing the concern about the choice of $\delta$. The added experiments using a $\delta$ appropriate for user-level differential privacy help resolve the original concern about the validity of the privacy guarantees. That said, it remains important that the paper clearly states throughout whether the analysis and experiments assume user-level or record-level privacy, and that results under the appropriate $\delta$ setting are presented in the final revised version. I expect this revision to be handled appropriately and raise my score accordingly.

---

> > > > > ### Author Response · Authors · 2025-08-05
> > > > >
> > > > > We sincerely thank Reviewer vd4X for raising the rating. We will make sure to clearly state throughout the final version whether the analysis and experiments assume user-level or record-level privacy.

---

### Decision · Program_Chairs · 2025-09-17

**Decision:**

Accept (spotlight)

**Comment:**

## Metareview

This paper studies decentralized federated learning (FL) scenario where there is no central server and the communication between the nodes happens at random. While the notion of local differential privacy (LDP) can be used in this setting, it is overly pessimistic (since it assumes e.g. that all except one nodes collude). Thus, two notions that more realistically captures the privacy guarantees in this model have been proposed: Pairwise Network DP (PN-DP) [Cyffers et al., ICML 2024] and Secret-based Local DP (Sec-LDP) [Allouah et al., ICML 2024]. The main contribution of the current work is to give significantly tighter privacy analysis within these frameworks.

More specifically, the aforementioned previous work focuses mostly on analyzing their privacy analysis through Renyi DP (RDP). However, as is known in central DP, privacy analysis through RDP can be quite loose and the notion of $f$-DP, which more tightly characterizes the privacy loss, can significantly improves upon RDP in the centralized setting. The technical contribution of this work is to apply $f$-DP to the decentralized setting under PN-DP and Sec-LDP frameworks. This requires the authors to overcome several technical challenges that arises due to the random nature of the communication. At the end, the authors achieve privacy guarantees that (numerically) significantly improves upon the RDP guarantees.

## Strengths

- Decentralized FL is a natural and important setting. The paper gives a rigorous and more realistic privacy guarantee in this model.

- In the empirical evaluation, the method in this paper achieves significant improvements in terms of privacy budget (more than 2x improvements for $\\epsilon$ in most cases) over RDP-based methods. As expected, these also translate to improvements in model accuracy when trained using a fixed budget.

- From a technical perspective, the analysis requires subtly combining the communication pattern with $f$-DP analysis. The technique might be useful to future work on the topic.

## Weaknesses

- The current analysis is tied to random-walk-based communication and correlated noise protocols. It is unclear how general these techniques  can be applied. In a reply to reviewer PKwX, the authors addressed this by sketching how the core ideas of the analysis can still be applied to other scenarios.

## Recommendation
This paper makes significant advances toward providing tighter and more realistic differential privacy guarantees in decentralized federated learning. Due to this, we support acceptance of the paper.